

# Estimation of pre-industrial nitrous oxide emissions from the land biosphere

Rongting Xu[1], Hanqin Tian[1], Chaoqun Lu[2,1], Shufen Pan[1], Jian Chen[3,1], Jia Yang[1], Bowen Zhang[1]

[1] International Center for Climate and Global Change Research and School of Forestry and Wildlife Sciences, Auburn University, Auburn, AL 36849, USA

[2] Department of Ecology, Evolution, & Organismal Biology, Iowa State University, Ames, IA 50011, USA

[3] College of Sciences and Mathematics, Auburn University, Auburn, AL 36849, USA

*Correspondence to:* Hanqin Tian (tianhan@auburn.edu)

**Abstract.** To accurately assess how increased global nitrous oxide ($N_2O$) emission has affected the climate system requires a robust estimation of the pre-industrial $N_2O$ emissions since only the difference between current and pre-industrial emissions represents net drivers of anthropogenic climate change. However, large uncertainty exists in previous estimates of pre-industrial $N_2O$ emissions from the land biosphere, while pre-industrial $N_2O$ emissions at the finer scales such as regional, biome, or sector have not yet well quantified. In this study, we applied a process-based Dynamic Land Ecosystem Model (DLEM) to estimate the magnitude and spatial patterns of pre-industrial $N_2O$ fluxes at the biome-, continental-, and global-level as driven by multiple environmental factors. Uncertainties associated with key parameters were also evaluated. Our study indicates that the mean of the pre-industrial $N_2O$ emission was approximately 6.20 Tg N yr$^{-1}$, with an uncertainty range of 4.76 to 8.13 Tg N yr$^{-1}$. The estimated $N_2O$ emission varied significantly at spatial- and biome-levels. South America, Africa, and Southern Asia accounted for 34.12%, 23.85%, 18.93%, respectively, together contributing of 76.90% of global total emission. The tropics were identified as the major source of $N_2O$ released into the atmosphere, accounting for 64.66% of the total emission. Our multi-scale estimates with a reasonable uncertainty range provides



a robust reference for assessing the climate forcing of anthropogenic N$_2$O emission from the land
biosphere.

## 1 Introduction

Nitrous oxide (N$_2$O) acts as the third-most important greenhouse gas (GHG) after carbon dioxide (CO$_2$)
and methane, contributing to the current radiative forcing (Myhre et al., 2013). Nitrous oxide is also the
most long-lived reactant, resulting in the destruction of stratospheric ozone (Prather et al., 2015;
Ravishankara et al., 2009). The atmospheric concentration of N$_2$O increased from 275 to 329 parts per
billion (ppb) since the pre-industrial era until 2015 at a rate of approximately 0.26% per year, as a result
of human activities (Davidson, 2009; Forster et al., 2007; NOAA2006A). The human-induced N$_2$O
emissions together with methane emissions from the terrestrial biosphere have offset terrestrial CO$_2$ sink
and contributed a net warming effect on the climate system (Tian et al., 2016). In the contemporary period,
anthropogenic N$_2$O emissions are mainly caused by the expansion in agricultural land area and increase
in fertilizer application, as well as industrial activities, biomass burning and indirect emissions from
reactive nitrogen (N) (Galloway et al., 2004; Reay et al., 2012). Natural terrestrial ecosystems contribute
more than half of N$_2$O released into the atmosphere when removing oceanic contribution (Denman et al.,
2007). As some N$_2$O emissions were present during pre-industrial times, only the difference between
current and pre-industrial emissions represents net drivers of anthropogenic climate change (Tian et al.,
2016). Therefore, it is necessary to provide a robust reference of pre-industrial N$_2$O emission for assessing
the climate forcing of anthropogenic N$_2$O emission from the land biosphere.





Numerous studies have reported the sources and estimates of $N_2O$ emission since the pre-industrial
era (Davidson and Kanter, 2014; Galloway et al., 2004; Kroeze et al., 1999; Syakila and Kroeze, 2011).
According to the Intergovernmental Panel on Climate Change Guidelines (IPCC, 1997), the global $N_2O$
emission evaluated by Kroeze et al. (1999) is 11 (8–13) Tg N yr$^{-1}$ (Natural soils: 5.6–6.6 Tg N yr$^{-1}$,
Anthropogenic: 1.4 Tg N yr$^{-1}$), which is consistent with the estimation from global pre-agricultural $N_2O$
emissions in soils (6–7 Tg N yr$^{-1}$) (Bouwman et al., 1993). While taking into account the new emission
factor from the IPCC 2006 Guidelines (Denman et al., 2007), Syakila and Kroeze (2011) conducted an
updated estimate based on the study of Kroeze et al. (1999) and reported that the global pre-industrial
$N_2O$ emission is 11.6 Tg N yr$^{-1}$ (Anthropogenic: 1.1 Tg N yr$^{-1}$, Natural soils: 7 Tg N yr$^{-1}$). Based on the
IPCC AR5, Davidson and Kanter (2014) indicated that the central estimates of both top-down and bottom-
up approaches for pre-industrial natural emissions were in agreement at 11 (10–12) Tg N yr$^{-1}$, including
natural emission from soils at 6.6 (3.3–9.0) Tg N yr$^{-1}$ (Syakila and Kroeze, 2011). Although these previous
estimates intent to provide a baseline of pre-industrial $N_2O$ emission at global-level, information on pre-
industrial $N_2O$ emissions on fine resolutions such as biome-, sector- or country-, and regional-levels
remains unknown but needed for climate change mitigation.
Large uncertainties in the estimates of pre-industrial $N_2O$ emission could derive from different
approaches (i.e. top-down and bottom-up), as mentioned above. Nitrous oxide, as an important component
of the N cycle, is produced by biological processes such as denitrification and nitrification in terrestrial
and aquatic systems (Schmidt et al., 2004; Smith and Arah, 1990; Wrage et al., 2001). In order to
accurately estimate pre-industrial $N_2O$ emissions using the process-based Dynamic Land Ecosystem
Model (DLEM, Tian et al., 2010), uncertainties associated with key parameters, such as maximum



nitrification and denitrification rates, biological N fixation (BNF) rates, and the adsorption coefficient for
soil ammonium ($NH_4^+$) and nitrate ($NO_3^-$), were required to be considered in model simulation. Upper
and lower limits of these parameters were used to derive a range of pre-industrial $N_2O$ emissions from
terrestrial ecosystems.
In this study, the DLEM was used to simulate global $N_2O$ emission in the pre-industrial era at a
resolution of 0.5° × 0.5° latitude/longitude. Since there is no observational data of $N_2O$ emission in the
pre-industrial period, a simple atmospheric box model (one-box) was applied to validate the estimated
$N_2O$ emissions from DLEM simulations. We calculated trends of $N_2O$ concentrations during 1860−2006
through accounting for all possible $N_2O$ sources from land biosphere and marine ecosystems based on the
previous publications. Then, the observational atmospheric $N_2O$ concentrations from monitoring stations
during 1977−2006 were used to compare with the calculated concentrations from the one-box model.
Finally, our estimates at global- and biome-scales were compared with previous estimates.

## 2 Methodology

### 2.1 Model description

The DLEM is a highly integrated process-based ecosystem model, which combines biophysical
characteristics, plant physiological processes, biogeochemical cycles, vegetation dynamics and land use
to make daily, spatially-explicit estimates of carbon, nitrogen and water fluxes and pool sizes in terrestrial
ecosystems from site- and regional- to global-scales (Lu and Tian, 2013; Tian et al., 2012). The DLEM
is characterized of cohort structure, multiple soil layer processes, coupled carbon, water and nitrogen
cycles, multiple GHG emissions simulation, enhanced land surface processes, and dynamic linkages





between terrestrial and riverine ecosystems (Liu et al., 2013; Tian et al., 2015; Tian et al., 2010). The
previous results of GHG emissions from DLEM simulations have been validated against field
observations and measurements at various sites (Lu and Tian, 2013; Ren et al., 2011; Tian et al., 2010;
Tian et al., 2011; Zhang et al., 2016). The estimates of water, carbon, and nutrients fluxes and storages
were also compared with the estimates from different approaches at regional-, continental-, and global-
scales (Pan et al., 2014; Tian et al., 2015; Yang et al., 2015).  Different soil organic pools and calculations
of decomposition rates were described in Tian et al. (2015). The decomposition and nitrogen
mineralization processes in the DLEM were described in other publications (Lu and Tian, 2013; Yang et
al., 2015).
**The N₂O module**
Previous work provided a detailed description of trace gas modules in the DLEM (Tian et al., 2010).
However, both denitrification and nitrification processes have been modified based on the first-order
kinetics (Chatskikh et al., 2005; Heinen, 2006).

In the DLEM, the N₂O production and fluxes are determined by soil inorganic N content ($NH_4^+$ and

$NO_3^-$) and environmental factors, such as soil texture, temperature, and moisture:
$$F_{\text{N2O}} = (R_{\text{nit}} + R_{\text{den}})F(T_{\text{soil}})(1 - F(Q_{\text{wfp}}))  \tag{1}$$
where $F_{\text{N2O}}$ is the N₂O flux from soils to the atmosphere (g N m² d⁻¹), $R_{\text{nit}}$ is the daily nitrification rate (g
N m² d⁻¹), $R_{\text{den}}$ is the daily denitrification rate (g N m² d⁻¹), $F(T_{\text{soil}})$ is the function of daily soil temperature
on nitrification process (unitless), and $F(Q_{\text{wfp}})$ is the function of water-filled porosity (unitless).



Nitrification, a process converting $NH_4^+$ into $NO_3^-$, is simulated as a function of soil temperature,
moisture, and soil $NH_4^+$ concentration:
$$R_{nit} = k_{nit}F(T_{soil})F(\psi)C_{NH_4} \qquad (2)$$
where $k_{nit}$ is the daily maximum fraction of $NH_4^+$ that is converted into $NO_3^-$ or gases (d$^{-1}$), $F(\psi)$ is the
soil moisture effect (unitless), and $C_{NH_4}$ is the soil $NH_4^+$ content (g N m$^{-2}$). Unlike Chatskikh *et al* 2005,
who set $k_{nit}$ to 0.10 d$^{-1}$, it varies with different plant function types (PFTs) in the DLEM with a range of
0.04 to 0.15 d$^{-1}$. The detailed calculations of $F(T_{soil})$ and $F(\psi)$ were described in Pan et al. (2015) and
Yang et al. (2015).
Denitrification is the process that converts $NO_3^-$ into three types of gases, namely, nitric oxide, $N_2O$,
dinitrogen. The denitrification rate is simulated as a function of soil temperature, water-filled porosity,
and $NO_3^-$ concentration $C_{NO_3}$ (g N g$^{-1}$ soil):
$$R_{den} = \alpha F(T_{soil})F(Q_{wfp})F_N(C_{NO_3}) \qquad (3)$$
where $F_N(C_{NO_3})$ is the dependency of the denitrification rate on $NO_3^-$ concentration (unitless), and α is
the maximum denitrification rate (g N m$^{-2}$ d$^{-1}$). The detailed calculations of $F(Q_{wfp})$, $F_N(C_{NO_3})$ and α were
described in Yang et al. (2015).
In each grid cell, there are four natural vegetation types and one crop type. The sum of $N_2O$ emission
in each grid/d$^{-1}$ is calculated by the following formula:
$$E = \sum_{i=1}^{62481}\sum_{j=1}^{5}\left(N_{ij} \times f_{ij}\right) \times A_i \times 10^6/10^{12}, \quad i = 1, \cdots, 62481, j = 1, \cdots, 5 \qquad (4)$$



where $E$ is the daily sum of $N_2O$ emission from all plant functional types (PFTs) in total grids (Tg N/yr$^{-1}$
d$^{-1}$); $N_{ij}$ (g N/m$^2$) is the $N_2O$ emission in the grid cell $i$ for PFT $j$; $f_{ij}$ is the fraction of cell used for PFT $j$
in grid cell $i$; and $A_i$ (km$^2$) is the area of the $i$th grid cell. $10^6$ is to convert km$^2$ to m$^2$ and $10^{12}$ is to convert
g to Tg.
**2.2 Input datasets**
Input data to drive DLEM simulation include static and transient data (Tian et al., 2010). Several
additional data sets were generated to better represent terrestrial environment in the pre-industrial period
as described below. The natural vegetation map was developed based on LUH (Hurtt et al., 2011) and
SYNMAP (Jung et al., 2006), which rendered the fractions of 47 vegetation types in each 0.5° grid. These
47 vegetation types were converted to 15 PFTs used in the DLEM through a cross-walk table (Figure 1).
Cropland distribution in 1860 were developed by aggregating the 5-arc minute resolution HYDE v3.1
global cropland distribution data (Figure 2). Half degree daily climate data (including average, maximum,
minimum air temperature, precipitation, relative humidity, and shortwave radiation) were derived from
CRU-NCEP climate forcing data (Wei et al., 2014). As global climate dataset was not available prior to
the year 1900, long-term average climate datasets from 1901 to 1930 were used to represent the initial
climate state in 1860. The nitrogen deposition dataset was developed based on the atmospheric chemistry
transport model (Dentener, 2006) constrained by the EDGAR-HYDE nitrogen emission data (Aardenne
et al., 2001). The nitrogen deposition dataset provided inter-annual variations of $NH_x$-N and $NO_y$-N
deposition rates. The manure production dataset (1961−2013) was derived from Food and Agriculture
organization of the United Nations statistic website ((FAO), http://faostat.fao.org) and defaulted for N





excretion rate referred to IPCC Guidelines (Zhang et al., in preparation). Estimates of manure production
from 1860 to 1960 were retrieved from the global estimates in (Holland et al., 2005).

**2.3 Model simulation**

The implementation of the DLEM simulation includes three steps: (1) equilibrium run, (2) spinning-up
run, and (3) transient run. In this study, we first used land use and land cover (LULC) map in 1860, long-
term mean climate during 1901−1930, N input datasets in 1860 (the concentration levels of N deposition
and manure application rate), and atmospheric $CO_2$ in 1860 to run the model to an equilibrium state. In
each grid, the equilibrium state was assumed to be reached when the inner-annual variations of carbon,
nitrogen, and water storage are less than 0.1 g $C/m^2$, 0.1 g $N/m^2$ and 0.1 mm, respectively, during two
consecutive 50 years. After the model reached equilibrium state, the model was spun up by the detrended
climate data from 1901 to 1930 to eliminate system fluctuation caused by the model mode shift from the
equilibrium to transient run (i.e., 3 spins with 10-year climate data each time). Finally, the model was run
in the transient mode with daily climate data, annual $CO_2$ concentration, manure application, and N
deposition inputs in 1860 to simulate pre-industrial $N_2O$ emissions. Additional description of model
initialization and simulation procedure can be found in previous publications (Tian et al., 2010).

**2.4 Model validation**

**2.4.1 Comparison with field measurements**

Observations of annual $N_2O$ emission accumulations (g N $m^{-2}$ $yr^{-1}$) were selected to compare with the
simulated emissions in different sites. As there were no field measurements in the pre-industrial era,
observations during 1970−2009 were collected to test the model performance in the contemporary period.



All environmental factors (climate, $CO_2$ concentration, soil property, N deposition, LULC) in the exact
year were used as input datasets for $N_2O$ simulations. The selected sites include temperate forest, tropical
forest, boreal forest, savanna, and grassland globally. As shown in Figure 3, the simulated $N_2O$ emissions
have a good correlation with field observations ($R^2 = 0.79$). It indicates that the DLEM has capacity to
simulate $N_2O$ emissions in the pre-industrial era driven by environmental factors back then. The detailed
information at each site can be found in Table 1S.
**2.4.2 One-box model validation**
A one-box model was used to estimate the accuracy of $N_2O$ fluxes from DLEM simulations (Kroeze et
al., 1999). The model equation is as follows:
$$\mathrm{d}C/\mathrm{d}t \; = \; S/F \; - \; C/T \qquad\qquad (5)$$
where $C$ is concentration (ppb), $S$ is emissions (Tg N), $T$ is atmospheric lifetime (years), $t$ is time (years),
and $F$ conversion factor (Tg N ppb$^{-1}$).

Atmospheric $N_2O$ concentration in 1860 derived from the records of Antarctic ice core was about 275

(263−280) ppb (Machida et al., 1995; Prather et al., 2012; Rahn and Wahlen, 2000; Spahni et al., 2005).
The atmospheric $N_2O$ concentration in 2006 was measured as about 320 ppb, which is approximately
18−20% higher than its pre-industrial value (Ciais et al., 2014). The atmospheric lifetime of $N_2O$ was 114
years, with a range of 106 to 141 years (Ciais et al., 2014; Prather et al., 2012; Prather and Hsu, 2010;
Volk et al., 1997).

The initial $N_2O$ concentration in the one-box atmospheric model was set as 275 ppb. $F$ conversion

factor is 4.8 Tg N ppb$^{-1}$ adopted from Kroeze et al. (1999). The atmospheric lifetime of $N_2O$ was set as





114 years. The mean with 95% confidence intervals, the maximum, and minimum values of estimates
from DLEM simulations were applied as initial emissions to calculate the atmospheric $N_2O$ concentration
in 2006 as shown in Table 1 (Scenarios 1–4 and baseline), as well as concentration changes from 1860 to
2006, as shown in Figure 7. According to the NOAA2006A, the monthly records of atmospheric $N_2O$
concentrations from different monitoring stations globally were from 1977 to 2015. Thus, the observed
trends from three stations: Pt. Barrow, Alaska, USA (71.3N, 156.6W), Mauna Loa, Hawaii, USA (19.5N,
155.6W), and South Pole (90S), were used to compare the calculated trends from all the above scenarios
during 1977 to 2006 (Figure 7). As uncertainties exist in the $N_2O$ concentration from ice core records and
the determination of its lifetime, the minimum and maximum estimates of them were used to calculate
the ranges of $N_2O$ concentrations in 2006, as shown in Table 1 (Scenarios 5–6).
**2.5 Estimate of uncertainty**
In this study, uncertainties in the simulated $N_2O$ emission were evaluated through a global sensitivity and
uncertainty analysis (Tian et al., 2011). Based on sensitivity analyses of key parameters that affect
terrestrial $N_2O$ fluxes, the most sensitive parameters were identified to conduct uncertainty simulations
in the DLEM, such as potential denitrification and nitrification rates, BNF rates, and the adsorption
coefficient for soil $NH_4^+$ and $NO_3^-$ (Gerber et al., 2010; Tian et al., 2015;Yang et al., 2015). The ranges
of five parameters were obtained from previous studies. Chatskikh et al. (2005) set $k_{nit}$ as 0.10 $d^{-1}$; however,
it was set in a range of 0.04 to 0.15 $d^{-1}$, and varied with different PFTs in the DLEM simulations. The
uncertainty ranges of potential nitrification rates were based on previous studies (Hansen, 2002; Heinen,
2006); the global pre-industrial N fixation was estimated as 58 Tg N $yr^{-1}$, ranging from 50–100 Tg N $yr^{-1}$
$^{1}$ (Vitousek et al., 2013). The spatial distribution of BNF referred to estimates done by Cleveland et al.



(1999). Potential denitrification rate was set in an uncertainty range of 0.025–0.74 $d^{-1}$, and varied with
different PFTs in the DLEM. The uncertainty ranges of the adsorption coefficient were referred to the
sensitivity analysis conducted in Yang et al. (2015). Parameters used in the DLEM simulations for
uncertainty analysis were assumed to follow a normal distribution. The Improved Latin Hypercube
Sampling (LHS) approach was used to randomly select an ensemble of 100 sets of parameters (R version
3.2.1) (Tian et al., 2015; Tian et al., 2011).
In the DLEM, after the model reached equilibrium state, a spinning-up run was implemented using
de-trended climate data from 1901 to 1930 for each set of parameter values. Then, each set of the model
was run in transient mode in 1860 to produce the result of the pre-industrial $N_2O$ emissions. All results
from 100 groups of simulations are shown in the Table 2S. The Shapiro–Wilk test was used on 100 sets
of results to check the normality of DLEM simulations. It turned out that the distribution is not normal (P
value $< 0.05$, R version 3.2.1), as shown in Figure 1S. Thus, the uncertainty range was represented as the
minimum and maximum value of 100 sets of DLEM simulations. In order to speculate the distribution of
the global mean $N_2O$ emission, we conduct the replicated Bootstrap resampling method (Efron and
Tibshirani, 1994) using 100 sets of DLEM simulation results. The 95% confidence intervals were
constructed with 10,000 replicates for defining the uncertainty bounds of the estimates of the global mean
$N_2O$ emission (Figure 2S).

## 220 3 Results & discussion

### 221 3.1 Magnitude and spatial distribution of $N_2O$ emission



We define the parameter-induced uncertainty of our global estimates as a range between the minimum
(4.76 Tg N yr$^{-1}$) and the maximum (8.13 Tg N yr$^{-1}$) of 100 sets of DLEM simulations. The global mean
N$_2$O emission was 6.20 Tg N yr$^{-1}$, with 95% confidence intervals of 6.03 to 6.36 Tg N yr$^{-1}$. The terrestrial
ecosystem in the pre-industrial period acted as a source of N$_2$O, and its spatial pattern mostly depends on
the biome distribution across the global land surface. The spatial distribution of annual N$_2$O emission in
a 0.5° × 0.5° grid (Figure 4) shows that the strong sources were found near the equator, such as Southeast
Asia, Central Africa, and Central America, where N$_2$O emission reached as high as 0.45 g N m$^{-2}$ yr$^{-1}$. The
weak N$_2$O sources were observed in the northern areas of North America and Asia, where the estimated
N$_2$O emission was less than 0.001 g N m$^{-2}$ yr$^{-1}$. The microbial activity in soils determined the rate of
nitrification and denitrification processes, which accounts for approximately 70% of global N$_2$O
emissions (Smith and Arah, 1990; Syakila and Kroeze, 2011). The tropical regions near the equator could
provide microbes optimum temperatures and soil moistures to decompose soil organic matter and release
more NO$_x$ and CO$_2$ into the atmosphere (Butterbach-Bahl et al., 2013). Referring to the observational data
from field experiments and model simulations in the tropics, it has been supported that the tropics are the
main sources within the total N$_2$O emissions from natural vegetation (Bouwman et al., 1995; Werner et
al., 2007; Zhuang et al., 2012).
In this study, Asia is divided into two parts: Southern Asia and Northern Asia, where the PFTs and
climate conditions are significantly contrasting. As shown in Figure 1, tropical forest and cropland were
dominant PFTs in Southern Asia. In contrast, temperate and boreal forests were main PFTs in Northern
Asia. The estimates of N$_2$O emissions from seven land regions are shown in Figure 5. At continental
scales, the N$_2$O emission was 2.09 (1.63–2.73) Tg N yr$^{-1}$ in South America, 1.46 (1.13–1.91) Tg N yr$^{-1}$



in Africa, and 1.16 (0.90–1.52) Tg N yr$^{-1}$ in Southern Asia. South America, Africa, and Southern Asia
accounted for 33.77%, 23.60%, 18.73%, respectively, together which was 76.10% of global total emission.
Europe and Northern Asia contributed to 0.45 (0.32–0.66) Tg N yr$^{-1}$, which was less than 10% of the total
emission.

Nitrous oxide emissions varied remarkably among different ecosystems. Forest, grassland, shrub,

tundra and cropland contributed 76.90%, 3.11%, 13.14%, 0.18% and 6.67%, respectively, to the total
emission globally (Figure 6). In different biomes, the tropics accounted for more than half of the total
$N_2O$ emission, which is comparable to the conclusion made by Bouwman et al. (1993). In the pre-
industrial era, the major inputs of reactive N to terrestrial ecosystems were from BNF, which relies on the
activity of a phylogenetically diverse list of bacteria, archaea and symbioses (Cleveland et al., 1999;
Vitousek et al., 2013). Tropical savannas have been considered as 'hot spots' of BNF by legume nodules
that provide the major input of available N (Bate and Gunton, 1982). The substantial inputs of N into
tropical forests could contribute to higher amount of the gaseous N losses as $N_2O$ or nitrogen gas
(Cleveland et al., 2010; Hall and Matson, 1999). In contrast, as the largest terrestrial biome, boreal forests
lack of available N because the rate of BNF is constricted by cold temperatures and low precipitation
during growing season (Alexander and Billington, 1986). Morse et al. (2015) conducted field experiments
in Northeastern North American forests. They found that denitrification does vary coherently with
patterns of N availability in forests, and no significant correlations between atmospheric N deposition,
potential net N mineralization and nitrification rates. Thus, it is reasonable that boreal forests contributed
to the least amount of $N_2O$ emission among different forests.



As shown in Figure 2, cropland areas varied spatially. The regions with high cropland area were the
entire Europe, India, eastern China, and central-eastern United States. The global $N_2O$ emission from
croplands was estimated as 0.41 (0.32–0.55) Tg N $yr^{-1}$, which is about ten times less than the estimate
reported in the IPCC AR5 (Ciais et al., 2014). As no synthetic N fertilizer was applied to the cropland in
1860, leguminous crops were the major source of $N_2O$ emission from croplands, most of which were
planted in central-eastern United States (Figure 4). Rochette et al. (2004) conducted the experiments on
the $N_2O$ emission from soybean without application of N fertilizer. Their work was in agreement with the
suggestion that legumes may increase $N_2O$ emissions compared with non-BNF crops (Duxbury et al.,
1982) The background emission from ground-based experiments was as high as 0.31–0.42 kg N $ha^{-1}$ in
Canada (Duxbury et al., 1982; Rochette et al., 2004).
We estimated pre-industrial $N_2O$ emissions from seventeen countries that are "hotspots" of $N_2O$
sources in the contemporary period (Table 2). The order of countries was referred to Gerber et al. (2016)
that indicated the top seventeen countries in terms of total N application in 2000. Pre-industrial $N_2O$
emissions from natural soils and croplands varied significantly at country-scales. The United States, China,
and India were top countries accounted for emissions from pre-industrial croplands. Countries close to or
located in the tropics, such as Mexico, Indonesia, and Brazil, accounted for negligible emissions from
croplands, but substantial amount from natural vegetation in the pre-industrial era. Previous studies
indicated that agriculture produces the majority of anthropogenic $N_2O$ emissions (Ciais et al., 2014;
Davidson and Kanter, 2014). Our estimate at country-scales could be used as a reference to quantify the
net increase of $N_2O$ emissions from agriculture activities in countries of "hotspots".



There is a debate that the natural wetlands and peatlands act as sinks or sources of $N_2O$. Previous
studies showed that $N_2O$ emissions from natural peatlands are usually negligible; however, the drained
peatlands with lower water tables might act as sources of $N_2O$ (Augustin et al., 1998; Martikainen et al.,
1993). High water tables in wetlands might block the activity of nitrifiers and limit the denitrification
(Bouwman et al., 1993). The fluxes of $N_2O$ were negligible in the pelagic regions of boreal ponds and
lakes due to the limitation of nitrification and/or nitrate inputs (Huttunen et al., 2003). Couwenberg et al.
(2011) mentioned that $N_2O$ emissions always decreased after rewetting when conducting field
experiments, which had been excluded from their future analysis of GHG emissions in peatlands. Hadi et
al. (2005) pointed out that tropical peatlands ranged from sources to sinks of $N_2O$, highly affected by
land-use and hydrological zone. In 1860, we were incapable to examine $N_2O$ fluxes from wetlands and
peatlands as human-induced land-use in those ecosystems was unknown. Thus, we excluded the $N_2O$
emissions from wetlands and peatlands in this study.
**3.2 Validation of DLEM results using the one-box model**
The sources of $N_2O$ include direct and indirect emissions. All anthropogenic emissions of $N_2O$ in 1860,
although in a low rate, were discussed in Davidson (2009), which included all direct emission from
biomass burning, fossil fuel combustion, etc. The net anthropogenic source in their work was estimated
as 0.42 Tg N $yr^{-1}$ in the pre-industrial period. However, the indirect emissions from the riverine induced
by the leaching and runoff of manure applications in agro-ecosystems, legume crop N fixation, and human
sewage discharging have not been addressed in Davidson (2009). According to the IPCC 1997, indirect
$N_2O$ emission was estimated as the total N leaching or runoff multiplied the emission factors. Through



combining the estimates from Davidson (2009) and emission factors from the IPCC 1997, the pre-
industrial indirect emission (Tg N yr⁻¹) was calculated as follows:

Indirect $N_2O$ emission = $0.3 \times (15 + 26.3 + 4.7) \times (0.015 + 0.0075 + 0.0025) = 0.35$

where 0.3 is the percentage of N through leaching or runoff (Sawamoto et al., 2005); 15, 26.3, and 4.7 Tg
N are the amount of crop fixed N, manure N, and human sewage N in the preindustrial era, respectively
(Davidson, 2009); 0.015, 0.0075, and 0.0025 are emission factors for degassing after discharge to surface
waters, in rivers, and in estuaries, respectively (IPCC 1997). Thus, the total emission from anthropogenic
activities in 1860 was estimated as 0.77 Tg N yr⁻¹, which was shown in Table 1. Syakila and Kroeze (2011)
assumed that $N_2O$ emission from oceans was 3.5 Tg N yr⁻¹, which had increased 1 Tg N yr⁻¹ since 1950
and was static at 4.5 Tg N yr⁻¹ from 2000–2006. In this study, $N_2O$ emission atmospheric sources were
assumed to be steady over time (Ciais et al., 2014). The net anthropogenic $N_2O$ emission in 2006 was
estimated as 7.2 Tg N yr⁻¹ (Syakila and Kroeze, 2011). Annual increase of net human-induced $N_2O$
emission was listed in Table 3S. All above possible sources of $N_2O$ emission in 1860 were used to
calculate the total emission, as listed in Table 1. The detailed calculation of the total emission in 1860 and
2006 can be found in the supplementary material.

As indicated by the calculated $N_2O$ concentration in 2006 for different scenarios (Table 1), the

estimated mean global $N_2O$ emission of 320.16 ppb was close to the observed concentrations in three
monitoring stations (MLO: 320.87; BRW: 320.73; SPO: 319.52 ppb) (NOAA2006A). However, the
increasing trends from monitoring stations and the one-box model calculations differed from each other.
The calculated increase rates of $N_2O$ concentrations from model calculation were higher than the observed



increase rates during 1977–1995. After the year 1995, the yearly increase rates from model calculations and observations were similar, as shown in Figure 7. The calculated concentration in 2006 from the upper and lower bound of the global mean is 318.02 and 322.30 ppb. The maximum concentration from the range of global mean emission was slightly lower than the calculation in Syakila and Kroeze (2011). It is because the initial and total emission (11.6; 19.8 Tg N yr$^{-1}$) from their study were higher than the estimates (11.23; 19.43 Tg N yr$^{-1}$) in this study. The calculated $N_2O$ concentrations in 2006 from scenario 3 is 304.61 ppb, which is much lower than the current concentration. Similarly, the result from scenario 4 is much higher than the observed $N_2O$ concentrations. Thus, we can conclude that the best estimate of $N_2O$ emission from pre-industrial global soils was around 6.20 (6.03–6.36) Tg N yr$^{-1}$. The extremely lower or higher estimates could not reflect the real $N_2O$ emission from terrestrial ecosystems under little human perturbation.

The uncertainty ranges in atmospheric lifetime and initial concentration could influence the calculation of atmospheric $N_2O$ concentration in 2006, as well as the trend of concentration changes since 1860. As shown in Table 1, lower lifetime resulted in the lower value of atmospheric $N_2O$ concentration, and vice versa. Similarly, lower initial atmospheric concentration resulted in lower estimate of atmospheric $N_2O$ concentration in 2006, and vice versa, while the effect is less significant than lifetime. Overall, we provide a reasonable estimation of $N_2O$ emission from the pre-industrial global soils in the context that the $N_2O$ concentration was 275 ppb and lifetime was set as 114 years.

**3.3 Comparison with other studies**



The global pre-agricultural $N_2O$ emission was estimated as 6.8 Tg N yr$^{-1}$ based on the regression
relationship between measured $N_2O$ fluxes and modeled $N_2O$ production indices (Bouwman et al., 1993).
This estimate was adopted to retrieve the trends of atmospheric $N_2O$ concentration in Syakila and Kroeze
(2011). In our study, the pre-industrial $N_2O$ emission from natural vegetation was estimated as 5.78
(4.4–7.72) Tg N yr$^{-1}$, which is about 1 Tg N yr$^{-1}$ lower than the estimate from Bouwman et al. (1993).
Estimate from the tropics (± 30° of the equator) was about 4.57 Tg N yr$^{-1}$, which is 0.83 Tg N yr$^{-1}$ lower
than the estimate from Bouwman et al. (1993). For the rest of natural vegetation, our estimate was 1.21
Tg N yr$^{-1}$, which is close to 1.4 Tg N yr$^{-1}$ estimated in Bouwman et al. (1993).

Although Bouwman et al. (1993) has studied the potential $N_2O$ emission from natural soils, our

study provided a first estimate of spatially distributed $N_2O$ emission in 1860 using the biogeochemical
process-based model. Bouwman et al. (1993) provided $1° \times 1°$ monthly $N_2O$ emission using the monthly
controlling factors without considering the impact of N deposition. In their study, the soil fertility and
carbon content were constant for every month, which could not reflect the monthly dynamic changes of
carbon and N pools in natural soils. Moreover, although their study has represented a spatial distribution
of potential $N_2O$ emission from natural soils, they had not provided the estimate at biome-, continent-,
and country-scales. Thus, their result was hardly to be used as a regional reference for the net human-
induced $N_2O$ emissions from some "hotspots", such as Southern Asia. In contrast, in our study, using
daily climate and N deposition dataset could better reflect the real variation of $N_2O$ emission through the
growing season in natural ecosystems. The comparison with field observations during 1997–2001
indicated that the DLEM can catch the daily peak $N_2O$ emissions in Hubbard Brook Forest (Tian et al.,
2010) and Inner-Mongolia (Tian et al., 2011).





As far as the N$_2$O emission from croplands, our estimate is comparable to the estimate of 0.3
(0.29–0.35) Tg N yr$^{-1}$ extracted from Syakila and Kroeze (2011) by digitizing graphs using the Getdata
Graph Digitizer (version 2.6.2, Russian Federation). In their study, the estimation was based on the
relationship between the crop production and human population during 1500–1970. In contrast, the result
in our study was estimated based on the cropland area of specific crop type, mainly soybean, rice, corn,
and wheat in 1860.
Thus, the DLEM is capable to provide the estimate of N$_2$O emission at regional- and biome-scales
with a higher spatial resolution, which could be a useful reference for studying how the LULC change,
such as tropical forest deforestation (Davidson, 2009), N fertilizer and manure application, and
increasingly atmospheric N deposition affect N$_2$O emissions in different terrestrial ecosystems or sectors
in the contemporary period.
**3.4 Future research needs**
Large uncertainty still exists in the DLEM simulation associated with the quality of input datasets and
parameters applied in simulations. Although input datasets could play a significant role in the variety of
the model output, it is difficult to obtain accurate datasets back to the year 1860. Average climate data
from 1901 to 1930 was used to run model simulation, which could raise the uncertainty in estimating N$_2$O
emission in 1860. The datasets of LULC, N deposition, and manure application in 1860 could introduce
uncertainties to this estimate. The estimates of human-induced N$_2$O emission could introduce the
uncertainty into the calculation of the N$_2$O concentrations in 2006. Nitrous oxide emission from inland
water system was calculated according to the empirical emission factor in the IPCC 1997. However, other



studies have indicated that the IPCC 1997 overestimated the indirect $N_2O$ emission (Hu et al., 2016;
Sawamoto et al., 2005). Thus, the estimate of indirect emission remains a large uncertainty. The $N_2O$
fluxes from wetlands and peats needed to be included in the future study.

## 386   4 Conclusions

Using the process-based land ecosystem model DLEM, this study provides a spatially-explicit estimate
of pre-industrial $N_2O$ emissions for major PFTs across global land surface. The one-box model was used
to calculate the atmospheric $N_2O$ concentration in 2006 to validate the results from DLEM simulations.
Improved LHS and Bootstrapping were performed to analyze uncertainty ranges of the estimates. We
estimated that pre-industrial $N_2O$ emission is 6.20 Tg N $yr^{-1}$. Calculated $N_2O$ concentration in 2006 using
the global mean $N_2O$ emission was 320.16 ppb, which was similar to the observed values from three
monitoring stations. The modeled results showed a large spatial variability due to variations in climate
conditions and PFTs. Tropical ecosystems were the dominant contributors of global $N_2O$ emissions. In
contrast, boreal regions contributed less than 5% to the total emission. China, India and United States are
top countries accounted for emissions from croplands in 1860. While uncertainties still exist in the $N_2O$
emission estimation in the pre-industrial era, this study offered a relatively reasonable estimate of the pre-
industrial $N_2O$ emission from land soils. Meanwhile, this study provided a spatial estimate for $N_2O$
emission from the global hotspots, which could be used as a reference to estimate net human-induced
emissions in the contemporary period.



## Author Contributions

*Xu R. performed DLEM simulations, analyses, calculations, and drafted the manuscript. Tian H. and Pan S. initiated this research and provided the comments for the whole work. Lu C. provided the idea of one-box model validation and contributed to the model calibration and data analysis. Chen J. contributed to the data processing and statistical analysis. Yang J. took charge of input datasets preparation (environmental factors), data description, and model verification. Zhang B. provided manure N input data and the comments on the manuscript.*

## Acknowledgements

*This work was supported by National Science Foundation (NSF) Grants (1243232, 121036). We wish to thank the previous members in the International Center for Climate and Global Change Research who made great contributions to the improvements of the DLEM in the past decade.*



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



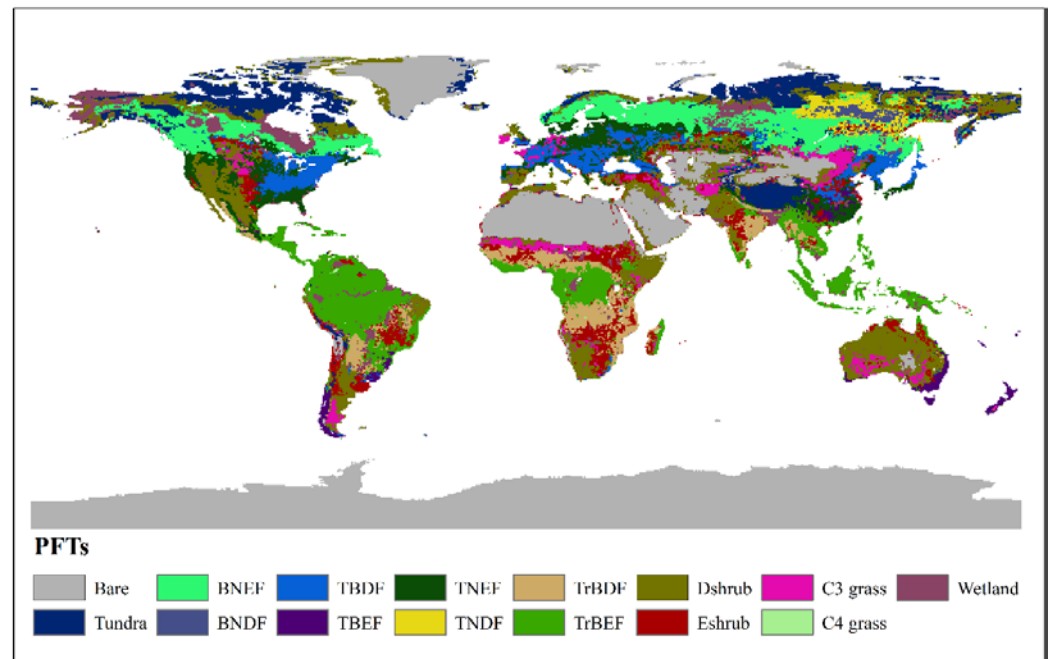

**Figure 1.** Global potential natural vegetation map used by DLEM in the pre-industrial era. BNEF: Boreal Needleleaf Evergreen Forest, BNDF: Boreal Needleleaf Deciduous Forest, TBDF: Temperate Broadleaf Deciduous Forest, TBEF: Temperate Broadleaf Evergreen Forest, TNEF: Temperate Needleleaf Evergreen Forest, TNDF: Temperate Needleleaf Deciduous Forest, TrBDF: Tropical Broadleaf Deciduous Forest, TrBEF: Tropical Broadleaf Evergreen Forest, Dshrub: Decidous Shrubland, Eshrub: Evergreen Shrubland.





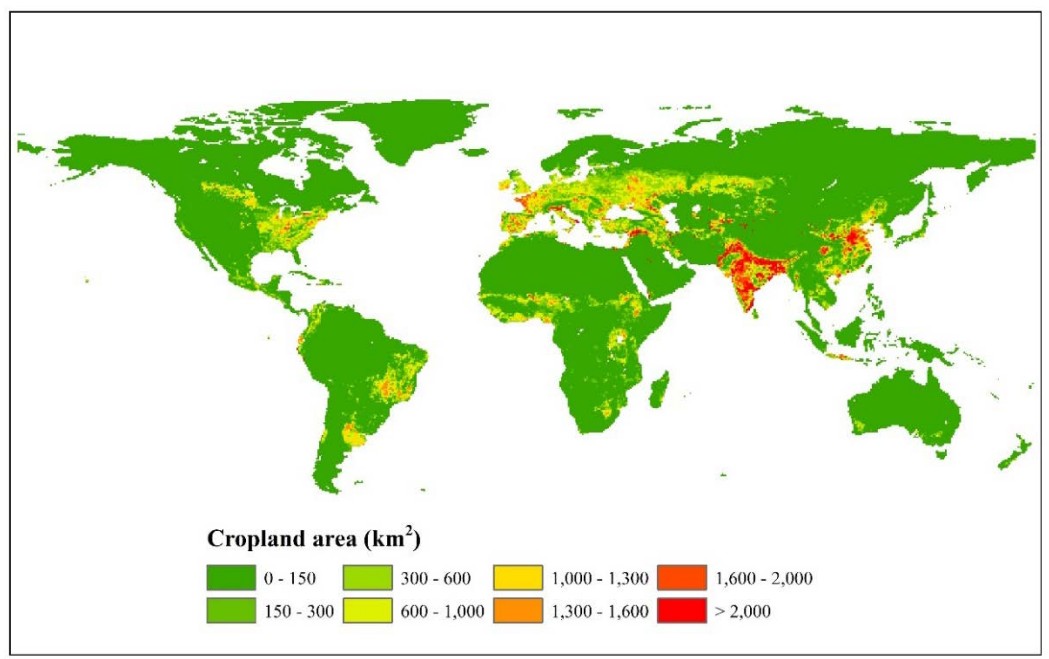

**Figure 2.** The spatial distribution of cropland area in 1860.

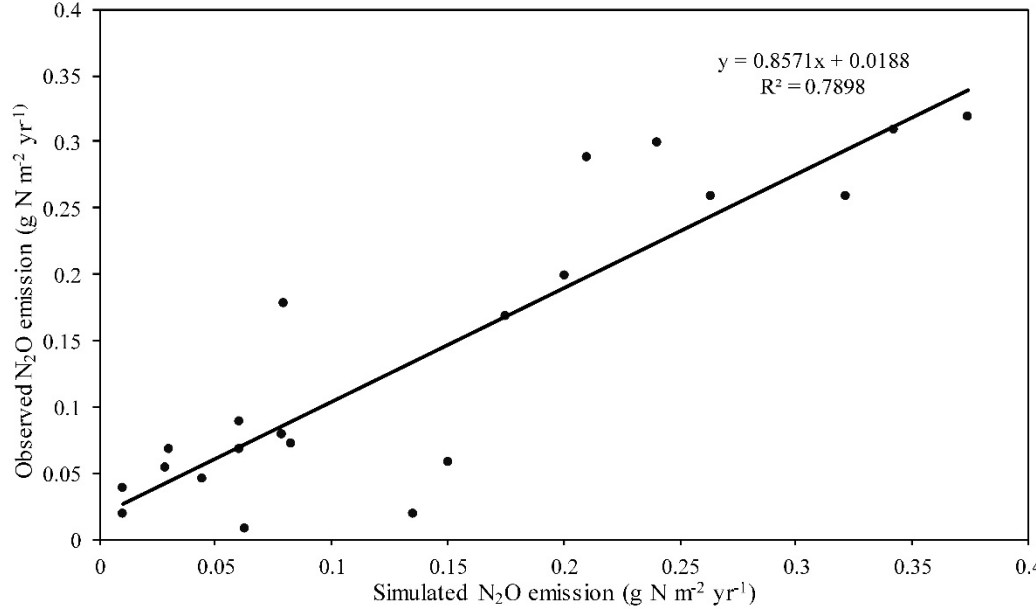

**Figure 3.** The comparison of the DLEM-simulated $N_2O$ emissions with field observations.



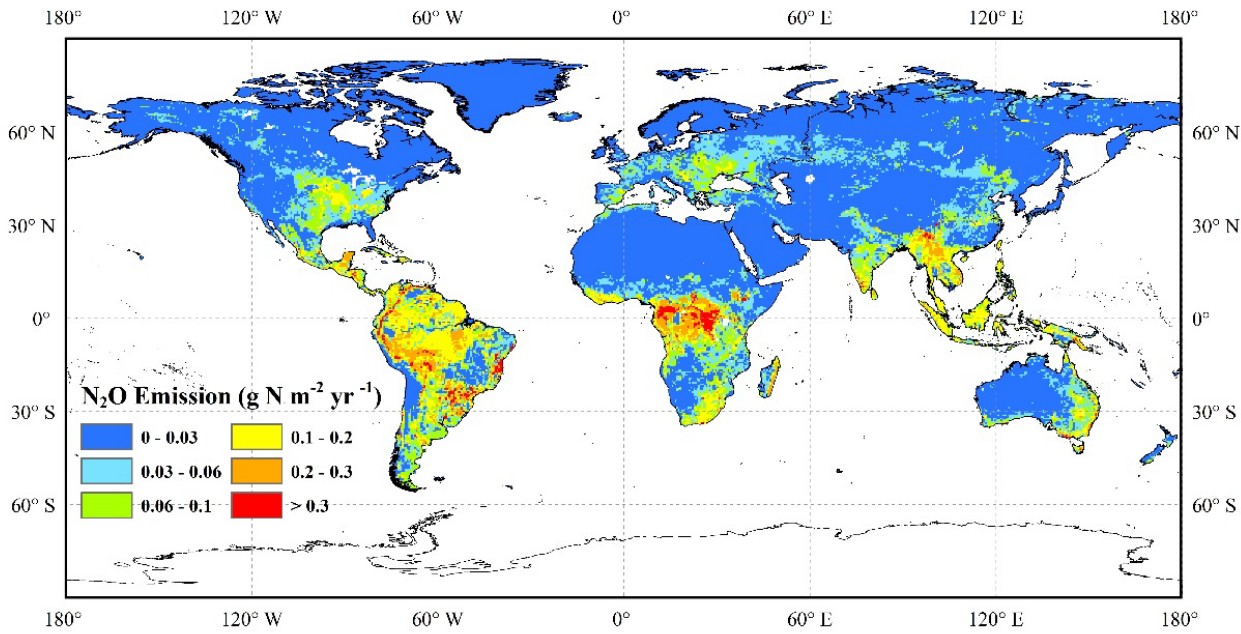

**Figure 4.** The spatial distribution of $N_2O$ emission in the pre-industrial era.



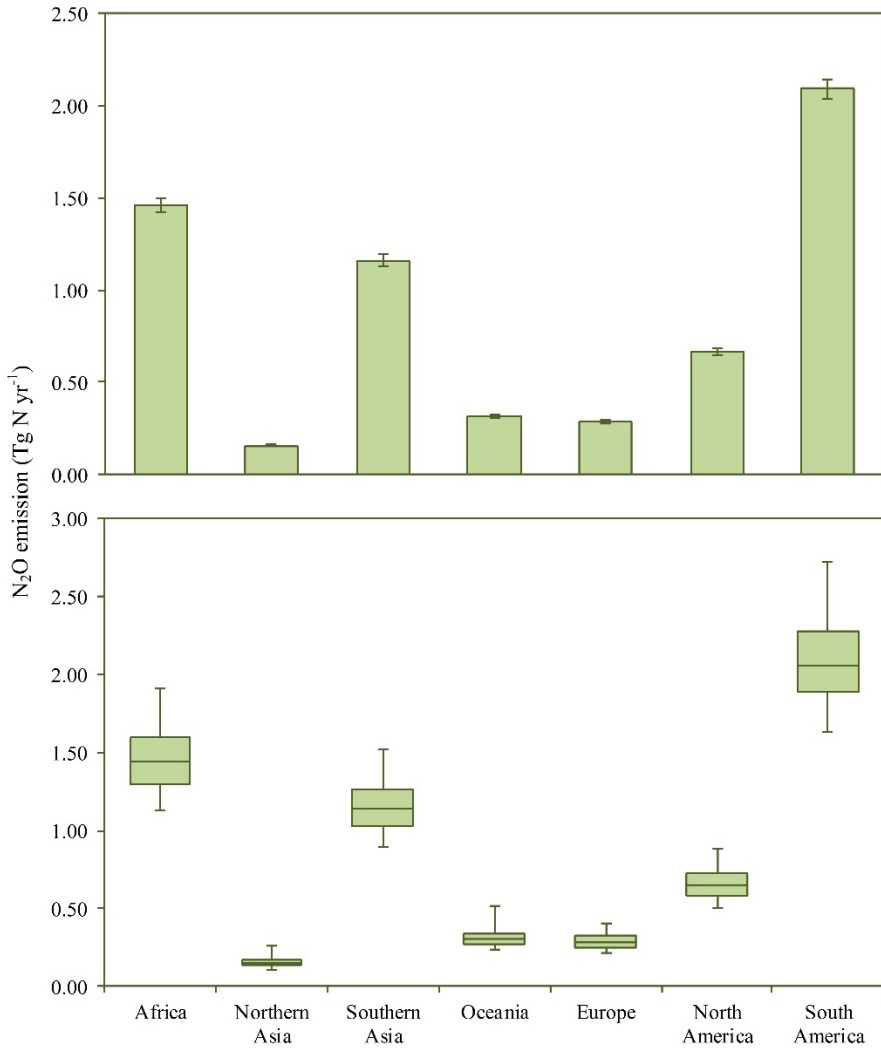

**Figure 5.** Estimated N$_2$O emissions at continental-level in 1860: the above graph is the mean emission from different continents with 95% confidence intervals; the below one is the median value and the uncertainty range of emissions.



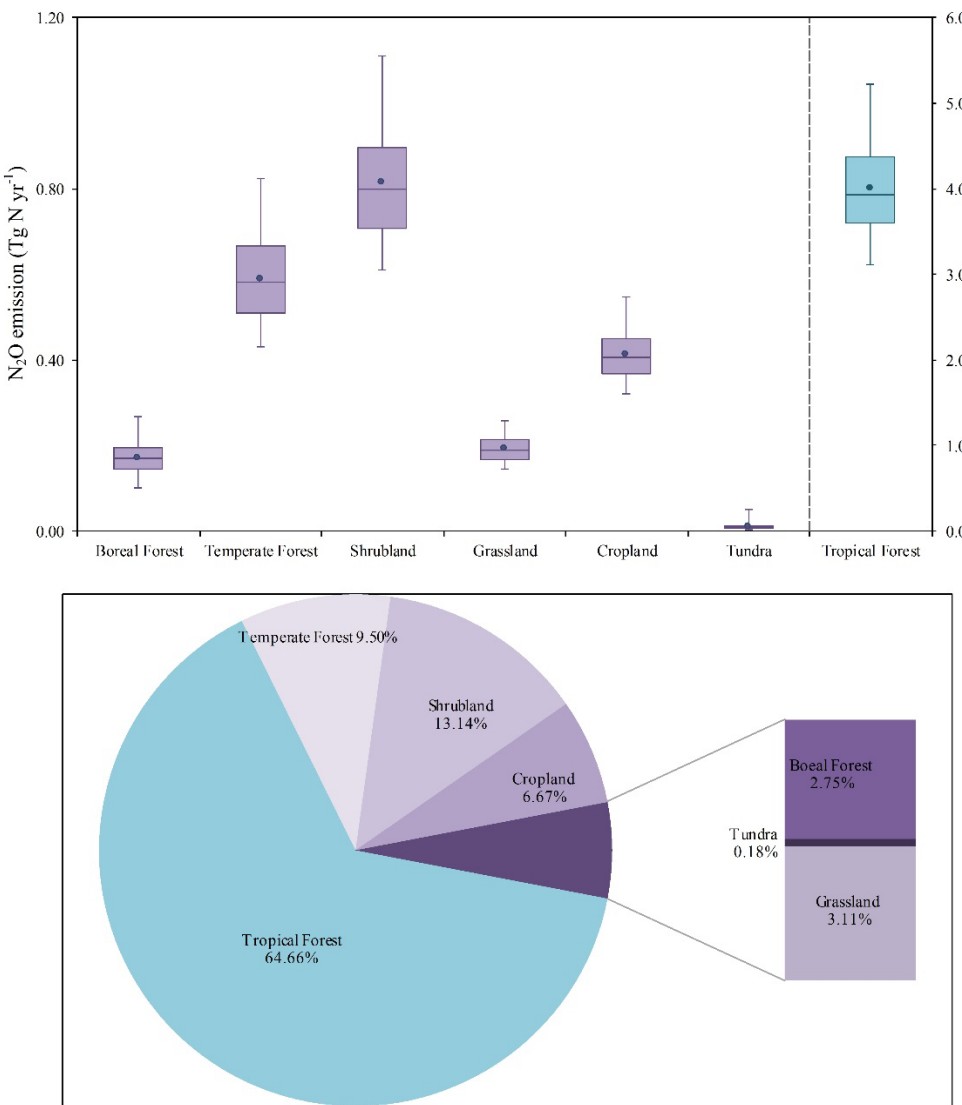

**Figure 6.** Estimated N$_2$O emissions at biome-level in 1860: the above graph is the median value (solid line), the mean (solid dot), and the uncertainty range of emissions from different biomes; the below one is the mean percentage of N$_2$O emissions.





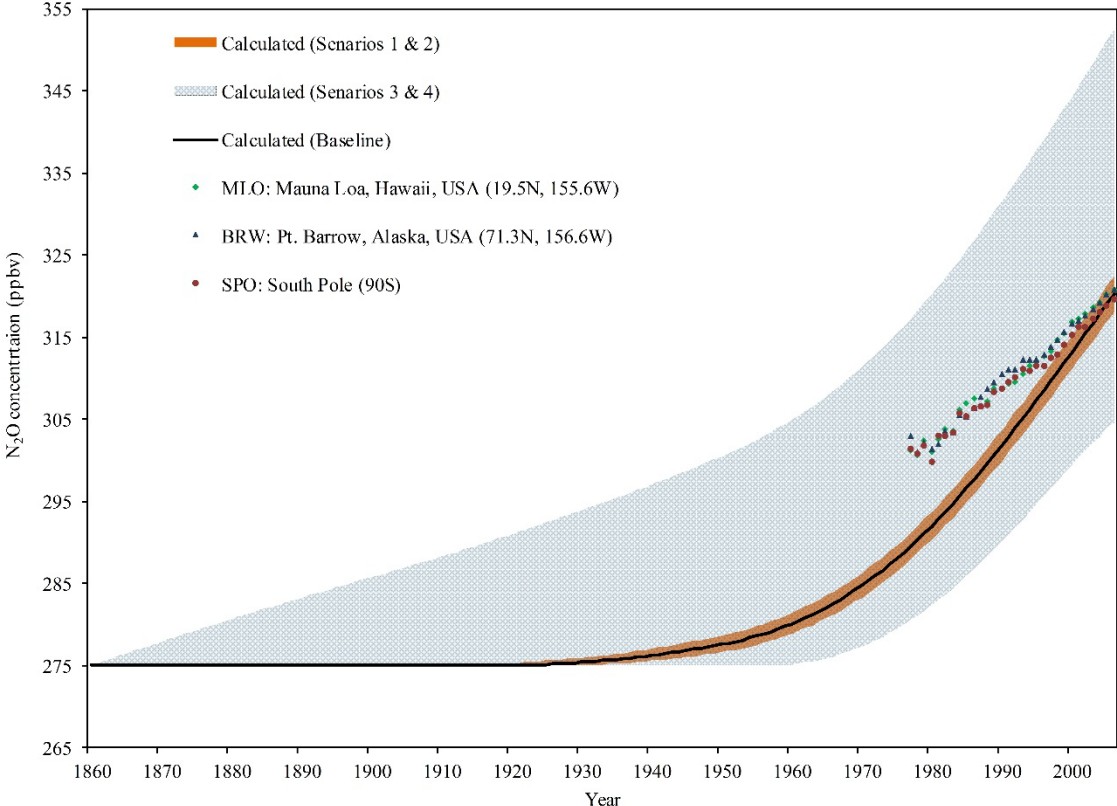

**Figure 7.** The trends of atmospheric concentration changes in different scenarios as described in Table 1. The method used to retrieve the trends of atmospheric $N_2O$ emission was directly adopted from Syakila and Kroeze (2011) and Kroeze et al. (1999). Similarly, annual emission was linearly interpolated between the years from 1860 to 2006 (net additions of anthropogenic $N_2O$ emission amount in different years were listed in Syakila and Kroeze, 2011). In this study, we focused on confirming the accuracy of pre-industrial estimates, as initial value, from our simulation instead of the accuracy of atmospheric trend itself as discussed in Syakila and Kroeze (2011).





**Table 1.** The estimate ranges of pre-industrial soil emissions, lifetime, and $N_2O$ concentration were used to calculate the concentrations of $N_2O$ in the atmosphere. Baseline was the mean estimate of $N_2O$ emissions from pre-industrial soils through DLEM simulation; Scenarios 1 & 2 were the lower bound and upper bound of the mean estimate, respectively; Scenarios 3 & 4 were the minimum and maximum estimates in this study, respectively; Scenarios 5.1 & 5.2 were the minimum and maximum estimates of $N_2O$ lifetime in the atmosphere, respectively; Scenarios 6.1 & 6.2 were the minimum and maximum estimates of atmospheric $N_2O$ concentration in 1860, respectively.

| Scenario | Terrestrial direct $N_2O$ emission (Tg N yr$^{-1}$) | Marine $N_2O$ emission (Tg N yr$^{-1}$) | Other sources (Tg N yr$^{-1}$) | Atmospheric chemistry (Tg N yr$^{-1}$) | Total emission (Tg N yr$^{-1}$) | $N_2O$ life time (years) | Atmospheric $N_2O$ concentration in 1860 (ppb) | Calculated atmospheric 2006 concentration (ppb) |
|---|---|---|---|---|---|---|---|---|
| Baseline | 6.20 | 3.5 (before 1950), 4.5 (after 1950) | 0.77 | 0.6 | 19.27 | 114 | 275 | 320.16 |
| Scenario1 | 6.03 | | | | 19.1 | | | 318.02 |
| Scenario2 | 6.36 | | | | 19.43 | | | 322.30 |
| Scenario3 | 4.76 | | | | 17.83 | | | 304.61 |
| Scenario4 | 8.13 | | | | 21.2 | | | 352.53 |
| Scenario5.1 | 6.20 | | | | 19.27 | 106 | 275 | 309.84 |
| Scenario5.2 | | | | | | 141 | | 362.15 |
| Scenario6.1 | | | | | | 114 | 263 | 314.21 |
| Scenario6.2 | | | | | | | 280 | 321.56 |



**Table 2.** Pre-industrial N$_2$O emissions from natural vegetation and croplands in different countries.

| Country | Vegetation area (Mha) | Natural soils (Gg N yr$^{-1}$) | Cropland (Gg N yr$^{-1}$) | Total (Gg N yr$^{-1}$) |
|---|---|---|---|---|
| China | 756.29 | 187.60 (Min: 143.13; Max:247.21) | 61.74 (Min: 46.65; Max: 83.43) | 249.34 (Min: 189.78; Max: 330.64) |
| India | 306.8 | 120.97 (Min:96.46; Max:153.91) | 64.29 (Min: 48.04; Max:86.86) | 185.26 (Min: 144.50; Max: 240.77) |
| United States | 913.93 | 296.45 (Min: 221.19; Max:409.61) | 80.95 (Min: 62.38; Max:106.29) | 377.40 (Min: 283.57; Max: 515.90) |
| Pakistan | 65.13 | 5.40 (Min: 4.05; Max:7.30) | 6.22 (Min: 4.90; Max:8.16) | 11.62 (Min: 8.95; Max: 15.46) |
| Indonesia | 174.07 | 181.35 (Min: 138.83; Max:237.96) | 1.98 (Min: 1.41; Max:2.97) | 183.33 (Min: 140.24; Max: 240.93) |
| France | 52.29 | 6.67 (Min: 4.77; Max:9.49) | 8.77 (Min: 6.60; Max:11.84) | 15.44 (Min: 11.37; Max: 21.33) |
| Brazil | 835.13 | 1016.53 (Min: 791.25; Max:1321.97) | 10.50 (Min: 8.00; Max:14.20) | 1027.03 (Min: 799.25; Max: 1336.17) |
| Canada | 914.61 | 93.96 (Min: 60.85; Max:137.00) | 2.47 (Min: 1.75; Max:3.49) | 96.43 (Min: 62.60; Max: 140.49) |
| Germany | 35.99 | 8.56 (Min: 6.22; Max:12.22) | 4.27 (Min: 3.18; Max:5.88) | 12.83 (Min: 9.40; Max: 18.10) |
| Turkey | 74.26 | 17.02 (Min: 12.37; Max:24.07) | 10.89 (Min: 8.42; Max:14.32) | 27.91 (Min: 20.79; Max: 38.39) |
| Mexico | 190.98 | 118.13 (Min: 90.56; Max:155.64) | 2.93 (Min: 2.43; Max:3.58) | 121.06 (Min: 92.99; Max: 159.22) |
| Vietnam | 31.71 | 41.38 (Min: 33.00; Max:53.27) | 2.15 (Min: 1.60; Max:2.84) | 43.53 (Min: 34.60; Max: 56.11) |
| Spain | 48.24 | 14.30 (Min: 10.70; Max:19.46) | 5.64 (Min: 4.23; Max:7.68) | 19.94 (Min: 14.93; Max: 27.14) |
| Russian Federation | 1575.27 | 233.98 (Min: 164.71; Max:333.24) | 19.28 (Min: 14.39; Max:26.20) | 253.26 (Min: 179.10; Max: 359.44) |
| Bangladesh | 12.41 | 1.61 (Min: 1.31; Max:2.02) | 5.38 (Min: 3.87; Max:7.59) | 6.99 (Min: 5.18; Max: 9.61) |
| Thailand | 49.26 | 55.87 (Min: 44.57; Max:71.77) | 2.62 (Min: 1.96; Max:3.56) | 58.49 (Min: 46.53; Max: 75.33) |

618