# Peer review of "Estimation of pre-industrial nitrous oxide emissions from the land biosphere"

_Climate of the Past, 2016_

## Short Comment (SC1) · 6 Nov 2016

General Comments

The objectives of this manuscript are to offer estimates of terrestrial sources of N2O emissions during pre-industrial (PI) times, both in terms of the total global sum and the spatial distribution of those emissions from soils across the continents. The complex process-based Dynamic Land Ecosystem Model 16 (DLEM) was employed, and appropriate driver datasets on land use and land cover, N deposition, climate, etc. were derived from literature sources to run the model. Overall, this is a useful exercise that has the potential to make a good contribution to the literature, but I do have several concerns regarding the approach and assumptions.

Specific Comments

1. With respect to constraints on the overall PI global emissions of N2O, I have more confidence in the top-down approach using atmospheric concentrations and lifetimes of N2O, than the bottom up simulations of a highly parameterized process model. The most recent top-down estimate (Prather et al., 2015) is cited in passing by the authors, but the estimates are not included in the present manuscript. The estimates from the IPCC AR4 and from Davidson & Kanter (2014), mentioned in lines 53-54, were based largely on the 2012 paper by Prather et al., but their 2015 paper provides an important update on lifetime estimates and resulting PI emission estimates. They now recommend using lifetimes of 123 years for PI and 116 years for the present (+/- 9 years), and from those lifetime estimates, they derive a new PI emission estimate of 10.5 Tg/yr. Fortunately, this is very close to other estimates, including the one from this study. Nevertheless, it should be specifically cited.

2. The point that the lifetime has probably decreased since PI times should be discussed. As far as I can tell, a varying lifetime cannot be incorporated into the one-box model (line 171) used by the authors. Perhaps the resulting global estimate is not terribly sensitive to this change, but that should be evaluated and discussed.

3. I fail to see how the analysis presented in Figure 7 and Table 1 provides additional confidence in the summed global estimate from this study. I can see the value of a sensitivity analysis of initial PI atmospheric concentrations and lifetimes, which Prather's papers have already done and for which they could be cited. In contrast, the analysis in Fig. 7 and Table 1 is clouded by the unclear source of annual emissions over the simulated time period and the validity of those assumptions. The text (lines 182-185) suggests that model output was used for annual emission estimates: "The mean with 95% confidence intervals, the maximum, and minimum values of estimates from DLEM simulations were applied as initial emissions to calculate the atmospheric N2O concentration in 2006 as shown in Table 1 (Scenarios 1−4 and baseline), as well as concentration changes from 1860 to 2006, as shown in Figure 7." However, the Fig. 7 captions indicates that the "net additions of anthropogenic N2O emission amount

in different years were listed in Syakila and Kroeze, 2011." I don't understand which was used to estimate annual increments of N2O concentration in Fig 7 – was it model output, as indicated on lines 182-185, or was it the net additions estimated by S&K as indicated in the figure caption? Both have problems.

S&K estimated fairly substantial N2O emissions from agriculture during the late 19th and early 20th centuries, but they also estimated a rather large decrease in natural emissions compared to 1500 (which are very difficult to estimate, see my further comments below), so their estimate of the net change relative to 1500 was small for this time period. However, the starting point for the present study is 1860. Therefore, it is incorrect to subtract this decline in natural emissions that preceded 1860 from the growth in anthropogenic emissions since 1860. S&K did this to show changes since their starting point of 1500, but using their "net additions" column without accounting for a different starting point in the present study introduces a significant bias. It is the net change relative to 1860 that is important for the present study, so the "net additions" estimated by S&K should be recalculated relative to 1860 if they are to be used in the analysis for Table 1 and Fig. 7.

I showed in my 2009 paper, and Smith et al. (2012) have affirmed, that atmospheric N2O began rising significantly many decades before fertilizer use became common in the 1950s, and so the "net additions" to the atmosphere must have been larger than those estimated by S&K relative to 1500, although they may be similar if they were corrected to be relative to 1860. We speculate that this increase in emissions between 1860 and 1950 was due to mineralization of soil N as agriculture expanded into regions of previously untilled soils, thus mobilizing N for rapid cycling, including a fraction lost at N2O. I also suspect that the current DLEM may not include effects of soil mining when virgin soil is first tilled, so if Table 1 is based on DLEM simulations, as indicated in the text on lines 182-185, then I suspect emissions from 1860 to 1950 were underestimated, which would affect the slope of the trend line later in the analysis as well.

I realize that the point of Figure 7 is not the accuracy of the simulated trend line, but rather the end point, but if the trend line agrees so poorly with the observations, then one has to question the validity of the model and the input data, which calls into question the reliability of the end point analysis. I believe that Fig. 7 and Table 1 could be replaced with citations of the sensitivity analyses done by Prather et al. (2012, 2015), but if the authors persist in wanting to include their own analysis, I would suggest that they utilize another source of "net addition" emissions than those of S&K relative to 1500.

4. The change in "natural" emissions before and after 1860 should be discussed. As I noted above, S&K deduce a substantial decline in natural emissions from 1500 to 1850. Similarly, I included a significant change in non-agricultural soil emissions due to tropical deforestation, which began growing rapidly in the late 20th century (Davidson 2009). Whether pre-1850 or post-1950, these changes in natural soil emissions are difficult to estimate, but the uncertainties that they represent should be considered, and biases resulting from how they are or are not included should be considered.

5. While the top-down approach of Prather et al. (2012, 2015) and the one box model used in the present study help constrain total PI emissions, the soil emission estimate must still be made by difference between total emissions and oceanic emissions. While the AR5 estimate of 3.8 Tg N2O-N/yr (range: 1.8 - 9.4; Ciais et al., 2013) is widely cited for emissions from the oceans, it is highly uncertain, so simply subtracting 3.8 (or 3.5 – 4.5 as in Table 1 of the present manuscript) from a total PI source estimate of about 11 Tg N2O-N/yr (+/- 1) doesn't really narrow the confidence estimate of the PI terrestrial source a great deal. Indeed, I just discovered a curious inconsistency between the AR5 best estimate of 3.8 with a review paper by Voss et al. (2013), which cites that same 3.8 value for N2O emissions from the open ocean, but then adds another 1.7 Tg N2O-N/yr for emissions from the continental shelf regions. I don't know if the AR5 review of the literature failed to adequately represent continental shelf regions or if Voss et al. might be double accounting. If Voss et al. are correct, the AR5 estimate of oceanic

emissions may be biased toward the low end, which would mean that the terrestrial PI source may more likely be in the range of 5 Tg N2O-N/yr or less. In any case, this highlights how uncertain the oceanic estimate is, which means we have to have similar uncertainty in the estimate of the PI terrestrial source. The narrow range of uncertainty in the present study's PI terrestrial source (6.03−6.36 Tg N2O-N/yr) reported on line 331 is unrealistically small.

6. The authors have misunderstood the emission estimates from my 2009 paper, which they incorrectly describe on lines 299-301: "However, the indirect emissions from the riverine induced by the leaching and runoff of manure applications in agro-ecosystems, legume crop N fixation, and human sewage discharging have not been addressed in Davidson (2009)." On the contrary, I derived emissions factors from a statistical model that was constrained by the historical record of atmospheric concentrations and fertilizer and manure use, so the emission factors derived from that analysis necessarily included all of the emissions, direct and indirect, that could be statistically correlated with historical fertilizer and manure use ("The sources attributed to fertilizers and manures include indirect emissions from downwind and downstream ecosystems, including human sewage." Davidson, 2009). Therefore, it is incorrect for the authors to calculate an additional indirect source (line 305) using IPCC default factors to add onto the estimate that they took from my paper that they misunderstood to be only direct emissions. They could either use an unmodified estimate from my paper or they could derive a new one, based on IPCC default values for both direct and indirect emissions based on estimates of BNF, fertilizer-N, and manure-N for 1860. Furthermore, note that the 0.42 Tg N2O-N/yr that they extracted from my paper for 1860 was for anthropogenic biological emissions (i.e., soils) only, and that there were also some other anthropogenic emissions at that time, such as biomass burning (see SI for Davidson 2009).

7. The authors should also acknowledge that there were anthropogenic effects on the N2O budget before 1860, so the 1860 fluxes don't necessarily represent only "natural" emissions. This includes some N2O from agricultural expansion that mined soil N and

also added BNF, some biomass burning, a tiny amount of industrial and transportation sector emissions, and possibly a loss of emissions from degraded natural soils that had been plowed for centuries or millennia, some of which were highly eroded.

8. Although my comments above all focus on the PI global total estimate, perhaps the more important contribution of this manuscript is the simulated spatial distribution of those PI soil emissions. It is not surprising that the model simulates the majority of the soil emissions coming from tropical forest soils. That is also true today for non-agricultural soils. There are a few curious details that jump out at me from the map (Fig. 4). Why are emissions from the Amazon Basin and SE Asia so much lower than from the Congo Basin? Other models that I am aware of don't show that difference (e.g., Zuang et al., 2012; Stehfest & Bouwman, 2006; Potter et al., 1996). Which of the datapoints in Fig. 3 are from tropical forests and which continents are they from? Is there validation support for the Congo having much higher emissions that the Amazon or SE Asia? More discussion would be helpful to interpret the variation shown in this map, such as where agriculture was or had been, where wetlands are, and where there are hot spots other than tropical forests. For example, I see a bunch of small red spots that appear to be near the Andes range, which puzzles me, but perhaps there is a good explanation. Ditto for why Northeastern Brazil, which is generally rather xeric, shows up as a hot spot. Also curious are the hot spots in southwestern China and the southeast coast of Australia.

Technical Points

line 41: This statement ignores that some anthropogenic emissions were already present prior to or at the beginning of the industrial revolution.

line 55: Add recent results from Prather et al. 2015.

line 70: Change "is" to "are" because the word "data" is plural: "the data are"

line 178: Use estimates from Prather et al. 2015.

line 312: Consider other estimates, such as those of Voss et al. 2013.

Figure 2. I don't understand the units. How can these units of crop area apply to each individual pixel?

Figure 3. The data used for this graph should be referenced.

Figure 5. The bottom panel is all that is needed. The top panel is redundant. However, you could also add a panel of mean flux per hectare, which would be useful, because it is difficult to compare fluxes across continents when the contents have such different total areas.

Figure 6. The two panels are largely redundant. The pie chart could include both the percentage of the total and the estimate of Tg/yr, which would obviate the need for the upper panel. However, again, the mean flux per hectare by biome would be an interesting panel to add.

Table 2. The number of significant figures shown is excessive. I suggest rounding to the nearest Gg. The uncertainties are such that any fraction of a Gg is meaningless.

Ciais, P., Sabine, C., Bala, G., Bopp, L., Brovkin, V., Canadell, J., Chhabra, A., DeFries, R., Galloway, J., Heimann, M. and Jones, C.: Carbon and other biogeochemical cycles. In Climate Change 2013: The Physical Science Basis. Contribution of Working Group I to the Fifth Assessment Report of the Intergovernmental Panel on Climate Change. Cambridge University Press, 465-570.

Davidson, E. A.: The contribution of manure and fertilizer nitrogen to atmospheric nitrous oxide since 1860, Nature Geoscience, 2, 659-662, 2009.

Davidson, E. A., and Kanter, D.: Inventories and scenarios of nitrous oxide emissions, Environmental Research Letters, 9, 105012, 2014.

Potter, C.S., Matson, P.A., Vitousek, P.M., and Davidson, E.A.: Process modeling of controls on nitrogen trace gas emissions from soils world-wide, J. Geophys. Res., 101,

1361-1377, 1996.

Prather, M. J., Holmes, C. D., and Hsu, J.: Reactive greenhouse gas scenarios: Systematic exploration of uncertainties and the role of atmospheric chemistry, Geophysical Research Letters, 39, 2012.

Prather, M. J., Hsu, J., DeLuca, N. M., Jackman, C. H., Oman, L. D., Douglass, A. R., Fleming, E. L., Strahan, S. E., Steenrod, S. D., and Søvde, O. A.: Measuring and modeling the lifetime of nitrous oxide including its variability, Journal of Geophysical Research: Atmospheres, 120, 5693-5705, 2015.

Smith, K.A., Mosier, A.R., Crutzen, P.J., and Winiwarter, W.: The role of N2O derived from crop-based biofuels, and from agriculture in general, in Earth's climate, Phil. Trans. R. Soc. B,367, 1169-1174, 2012.

Stehfest, E., and Bouwman, L.: N2O and NO emission from agricultural fields and soils under natural vegetation: summarizing available measurement data and modeling of global annual emissions, Nutrient Cycling in Agroecosystems, 74, 207 –228, 2006.

Syakila, A., and Kroeze, C.: The global nitrous oxide budget revisited, Greenhouse Gas Measurement and Management, 1, 17-26, 2011.

Voss, M, Bange, H.W., Dippner, J.W., Middelburg, J.J., Montoya, J.P., and Ward, B.: The marine nitrogen cycle: recent discoveries, uncertainties and the potential relevance of climate change, Phil. Trans. R. Soc. B,368, 20130121, 2013.

Zhuang, Q., Lu, Y., and Chen, M.: An inventory of global N2O emissions from the soils of natural terrestrial ecosystems, Atmospheric Environment, 47, 66-75, 2012.

Respectfully submitted,

Eric A. Davidson University of Maryland Center for Environmental Science, Appalachian Laboratory

---

## Referee Comment (RC2) · Anonymous Referee #2 · 27 Feb 2017

This is an interesting study that reports the pre-industrial nitrous oxide emissions from the land biosphere. The authors used a process-based ecosystem model, DLEM, to simulate the N2O emission based on assumed previous-industrial conditions of climate, vegetation cover etc. Several comments might be helpful for the authors to further improve the manuscript.

The country-level analysis does not make much sense as a large amount of countries had different boundaries compared to present. In line 396, those country-level emissions might need to be removed.

I am little curious to see the small uncertainties in continent-level N2O show in Figure 5 as the LHS was used and the large uncertainties were shown in below panel in Figure 5.

The model implementation is not clear. I assume this study is based on a steady state or semi-steady state simulation. The equilibrium run was for 1860, followed by a spinup. The transient run was driven with climate data in 1860 (line 153). What is the data source? If the equilibrium run was based on 1860 data (most). Then, there are small discrepancies among spinup and transient runs. A comparison between equilibrium and transient run might be needed. If there are no big differences, using equilibrium run might be more convincing, as most driving forces were 1860 except climate data of 1901-1930. If the authors really want to have a transient run, the model simulations should start even further to capture the legacy impacts of natural and anthropogenic impacts, particularly the land use change.
* * *

---

## Author Comment (AC1) · 26 Mar 2017

Dear reviewer #1,

Thanks very much for the precious comments and suggestions. All the comments and suggestions are addressed in the revised manuscript.

1. With respect to constraints on the overall PI global emissions of N2O, I have more confidence in the top-down approach using atmospheric concentrations and lifetimes of N2O, than the bottom up simulations of a highly parameterized process model. The most recent top-down estimate (Prather et al., 2015) is cited in passing by the authors, but the estimates are not included in the present manuscript. The estimates from the IPCC AR4 and from Davidson & Kanter (2014), mentioned in lines 53-54, were based largely on the 2012 paper by Prather et al., but their 2015 paper provides an

important update on lifetime estimates and resulting PI emission estimates. They now recommend using lifetimes of 123 years for PI and 116 years for the present (+/- 9 years), and from those lifetime estimates, they derive a new PI emission estimate of 10.5 Tg/yr. Fortunately, this is very close to other estimates, including the one from this study. Nevertheless, it should be specifically cited.

Response: We have cited the recent study by Prather et al. (2015) in the introduction and discussion sections of the revised manuscript.

Line 53-55: Prather et al. (2015) provided an estimate of the pre-industrial emissions (total natural emission: 10.5 Tg N yr-1) based on the most recent study with a corrected lifetime of 116 years.

Line 260-267: "Top-down" methodology used to estimate N2O emissions is based on atmospheric measurements and an inversion model (Thompson et al. 2014). Prather et al. (2012) provided an estimate of $9.1\pm1.0$ Tg N yr-1 of natural emission in the pre-industrial era using observed pre-industrial abundances of 270 ppb and model estimates of lifetime decreased from 142 years in the pre-industrial era to $131\pm10$ years in the present-day. Later, Prather et al. (2015) re-evaluated N2O lifetime based on Microwave Limb Sounder satellite measurements of stratospheric, which was consistent with modeled values in the present-day. The lifetime in the pre-industrial era and present-day was estimated as 123 and $116\pm9$ years, respectively. The current lifetime increases the pre-industrial natural emission from $9.1\pm1.0$ to 10.5 Tg N yr-1.

2. The point that the lifetime has probably decreased since PI times should be discussed. As far as I can tell, a varying lifetime cannot be incorporated into the one-box model (line 171) used by the authors. Perhaps the resulting global estimate is not terribly sensitive to this change, but that should be evaluated and discussed.

Response: In the revised version, we have removed the one-box model validation. In addition, we added the discussion of the decreased lifetime since PI times, which was also mentioned in the response to question #1 (Line 260-267).

3. I fail to see how the analysis presented in Figure 7 and Table 1 provides additional confidence in the summed global estimate from this study. I can see the value of a sensitivity analysis of initial PI atmospheric concentrations and lifetimes, which Prather's papers have already done and for which they could be cited. In contrast, the analysis in Fig. 7 and Table 1 is clouded by the unclear source of annual emissions over the simulated time period and the validity of those assumptions. The text (lines 182-185) suggests that model output was used for annual emission estimates: "The mean with 95% confidence intervals, the maximum, and minimum values of estimates from DLEM simulations were applied as initial emissions to calculate the atmospheric N2O concentration in 2006 as shown in Table 1 (Scenarios 1-4 and baseline), as well as concentration changes from 1860 to 2006, as shown in Figure 7." However, the Fig. 7 captions indicates that the "net additions of anthropogenic N2O emission amount in different years were listed in Syakila and Kroeze, 2011." I don't understand which was used to estimate annual increments of N2O concentration in Fig 7 – was it model output, as indicated on lines 182-185, or was it the net additions estimated by S&K as indicated in the figure caption? Both have problems. S&K estimated fairly substantial N2O emissions from agriculture during the late 19th and early 20th centuries, but they also estimated a rather large decrease in natural emissions compared to 1500 (which are very difficult to estimate, see my further comments below), so their estimate of the net change relative to 1500 was small for this time period. However, the starting point for the present study is 1860. Therefore, it is incorrect to subtract this decline in natural emissions that preceded 1860 from the growth in anthropogenic emissions since 1860. S&K did this to show changes since their starting point of 1500, but using their "net additions" column without accounting for a different starting point in the present study introduces a significant bias. It is the net change relative to 1860 that is important for the present study, so the "net additions" estimated by S&K should be recalculated relative to 1860 if they are to be used in the analysis for Table 1 and Fig. 7. I showed in my 2009 paper, and Smith et al. (2012) have affirmed, that atmospheric N2O began rising significantly many decades before fertilizer use became common in the 1950s,

and so the "net additions" to the atmosphere must have been larger than those esti­mated by S&K relative to 1500, although they may be similar if they were corrected to be relative to 1860. We speculate that this increase in emissions between 1860 and 1950 was due to mineralization of soil N as agriculture expanded into regions of previ­ously untilled soils, thus mobilizing N for rapid cycling, including a fraction lost at N2O. I also suspect that the current DLEM may not include effects of soil mining when virgin soil is first tilled, so if Table 1 is based on DLEM simulations, as indicated in the text on lines 182-185, then I suspect emissions from 1860 to 1950 were underestimated, which would affect the slope of the trend line later in the analysis as well. I realize that the point of Figure 7 is not the accuracy of the simulated trend line, but rather the end point, but if the trend line agrees so poorly with the observations, then one has to question the validity of the model and the input data, which calls into question the reliability of the end point analysis. I believe that Fig. 7 and Table 1 could be replaced with citations of the sensitivity analyses done by Prather et al. (2012, 2015), but if the authors persist in wanting to include their own analysis, I would suggest that they utilize another source of "net addition" emissions than those of S&K relative to 1500.

Response: According to your suggestions, we have removed the one-box model vali­dation. Instead, we cited the work done by Prather et al. (2012, 2015) and compared our results with theirs in the section 3.2.

Line 268-277: Natural sources for N2O include soil under natural vegetation, oceans, and atmospheric chemistry (Ciais et al., 2014). The emission from atmospheric chem­istry was estimated as 0.6 with an uncertainty range of 0.3-1.2 Tg N yr-1. Syakila and Kroeze (2011) estimated global natural emissions from oceans as 3.5 Tg N yr-1. Oceanic emission was estimated as 3.8 with an uncertainty range of 1.8-5.8 Tg N yr-1 in the IPCC AR4. However, the uncertainty range became larger (1.8-9.4 Tg N yr-1) in the IPCC AR5. In our study, the simulated N2O emission was from agri­cultural and natural soils. The natural emission was estimated as 5.78 (4.4-7.72) Tg N yr-1. Combining the atmospheric chemistry and the ocean emissions in the IPCC

AR5 with the natural emissions from our study, the global total natural N2O emissions were 10.18 (6.5-18.32) Tg N yr-1. The large uncertainty range was attributed to the uncertainty from oceanic emission, atmospheric chemistry emission, and our estimation. The estimated global total amount (10.18 Tg N yr-1) in this study was comparable to the estimate (10.5 Tg N yr-1) by Prather et al. (2015) using the top-down approach.

4. The change in "natural" emissions before and after 1860 should be discussed. As I noted above, S&K deduce a substantial decline in natural emissions from 1500 to 1850. Similarly, I included a significant change in non-agricultural soil emissions due to tropical deforestation, which began growing rapidly in the late 20th century (Davidson 2009). Whether pre-1850 or post-1950, these changes in natural soil emissions are difficult to estimate, but the uncertainties that they represent should be considered, and biases resulting from how they are or are not included should be considered.

Response: We agree that different factors caused different variation patterns in N2O fluxes before and after 1860. We did not consider the pre-1850 natural emission change because we assumed emission in 1860 can represent the pre-industrial level although it has declined from 1500 to 1850. Our estimation from the process-based model can capture the N2O emission due to land use change in the late 20th century, but it is beyond the scope of this paper. Since pre-industrial N2O emission is not always stable and remains a large uncertainty, our estimation can only go back to 1860 and represent N2O level before intensive human disturbance.

5. While the top-down approach of Prather et al. (2012, 2015) and the one box model used in the present study help constrain total PI emissions, the soil emission estimate must still be made by difference between total emissions and oceanic emissions. While the AR5 estimate of 3.8 Tg N2O-N/yr (range: 1.8 - 9.4; Ciais et al., 2013) is widely cited for emissions from the oceans, it is highly uncertain, so simply subtracting 3.8 (or 3.5 – 4.5 as in Table 1 of the present manuscript) from a total PI source estimate of about 11 Tg N2O-N/yr (+/-1) doesn't really narrow the confidence estimate of the PI terrestrial source a great deal. Indeed, I just discovered a curious inconsistency between the AR5

best estimate of 3.8 with a review paper by Voss et al. (2013), which cites that same 3.8 value for N2O emissions from the open ocean, but then adds another 1.7 Tg N2O-N/yr for emissions from the continental shelf regions. I don't know if the AR5 review of the literature failed to adequately represent continental shelf regions or if Voss et al. might be double accounting. If Voss et al. are correct, the AR5 estimate of oceanic emissions may be biased toward the low end, which would mean that the terrestrial PI source may more likely be in the range of 5 Tg N2O-N/yr or less. In any case, this highlights how uncertain the oceanic estimate is, which means we have to have similar uncertainty in the estimate of the PI terrestrial source. The narrow range of uncertainty in the present study's PI terrestrial source (6.03-6.36 Tg N2O-N/yr) reported on line 331 is unrealistically small.

Response: Yes, the soil emission estimation must still be made by difference between total emissions and oceanic emissions regardless of methodology (top-down or bottom-up). In the IPCC AR5, the average oceanic emission is 3.8 Tg N yr-1, with a larger uncertainty range compared with the estimate in the AR4. The estimate from Voss et al. (2013) indicated that oceanic emission was 1.7 Tg N yr-1 more than the average in the AR5. It is because they considered the emissions (1.7 Tg N yr-1) from "rivers, estuaries, and coastal zones" as the marine emissions, as written in Table 7.7 of the IPCC AR4 Chapter 7. Thus, the average estimation in AR5 is still trustable. In this study, to compare with the results (10.5 Tg N yr-1) in Prather et al. (2015), we need to sum our estimate and other natural emissions. The global total natural N2O emissions were 10.18 (6.5-18.32) Tg N yr-1 in the preindustrial era.

The small uncertainty range shown in the upper panel of Fig. 5 was the 95% confidence interval of the mean estimate, as explained in the manuscript. The uncertainty range of pre-industrial N2O emissions was present using the minimum and maximum estimate (4.76-8.13 Tg N yr-1) in this study, which was consistent with other studies, such as the reported estimates in the IPCC AR5. Here, the Bootstrap resampling method was used to define the uncertainty bounds of global mean N2O emission (6.20 Tg N yr-1)

(shown in line 216-219 of previous manuscript). It was used to verify the stability of the LHS approach. The 95% confidence intervals (6.03-6.36 Tg N yr-1) of the mean did not represent the uncertainty range for pre-industrial N2O emission in this study. Thus, we will not report this narrow range in the revised manuscript to avoid the confusion.

6. The authors have misunderstood the emission estimates from my 2009 paper, which they incorrectly describe on lines 299-301: "However, the indirect emissions from the riverine induced by the leaching and runoff of manure applications in agro-ecosystems, legume crop N fixation, and human sewage discharging have not been addressed in Davidson (2009)." On the contrary, I derived emissions factors from a statistical model that was constrained by the historical record of atmospheric concentrations and fertilizer and manure use, so the emission factors derived from that analysis necessarily included all of the emissions, direct and indirect, that could be statistically correlated with historical fertilizer and manure use ("The sources attributed to fertilizers and manures include indirect emissions from downwind and downstream ecosystems, including human sewage." Davidson, 2009). Therefore, it is incorrect for the authors to calculate an additional indirect source (line 305) using IPCC default factors to add onto the estimate that they took from my paper that they misunderstood to be only direct emissions. They could either use an unmodified estimate from my paper or they could derive a new one, based on IPCC default values for both direct and indirect emissions based on estimates of BNF, fertilizer-N, and manure-N for 1860. Furthermore, note that the 0.42 Tg N2O-N/yr that they extracted from my paper for 1860 was for anthropogenic biological emissions (i.e., soils) only, and that there were also some other anthropogenic emissions at that time, such as biomass burning (see SI for Davidson 2009).

Response: We are sorry for the misunderstanding of this paper. We deleted this sentence and recalculated the overall N2O emissions from this paper. In Davidson (2009), two approaches (top-down and bottom-up) had been applied to estimate the anthropogenic biogenic N2O emissions in 1860. The estimates from top-down and bottom-up were 0.42 and 0.54 Tg N yr-1, respectively. Thus, the final number we used in this

study was 0.5 with an uncertainty range of 0.4-0.6 Tg N yr-1. In addition, N2O from the biomass burning was assumed to be 0.2 Tg N yr-1 in 1860 in Davidson (2009). In sum, the total anthropogenic N2O emission in 1860 was estimated as 0.7 (0.6-0.8) Tg N yr-1 in Davidson (2009). We added below content in the revised version:

Line 321-323: The pre-industrial anthropogenic N2O sources in his study included biomass burning, agriculture activities (e.g., manure and fertilizer application, and the cultivation of legume) and human sewage, the sum of which was 0.7(0.6-0.8) Tg N yr-1 (Davidson, 2009).

7. The authors should also acknowledge that there were anthropogenic effects on the N2O budget before 1860, so the 1860 fluxes don't necessarily represent only "natural" emissions. This includes some N2O from agricultural expansion that mined soil N and also added BNF, some biomass burning, a tiny amount of industrial and transportation sector emissions, and possibly a loss of emissions from degraded natural soils that had been plowed for centuries or millennia, some of which were highly eroded.

Response: Yes, the 1860 fluxes don't necessarily represent only "natural" emissions. In our study, when we mentioned "natural" emissions, we excluded the emissions from cropland soils. Our study has only addressed the anthropogenic emissions from cropland expansion and manure application, but we are unable to simulate the anthropogenic emissions from biomass burning and other sectors. As described in the response 6, we have added the discussion on the pre-industrial anthropogenic N2O emission in this manuscript (Section 3.4 Line 321-323).

See the section 3.4, Line 311-334:

3.4 The N2O budget in the pre-industrial era The observed N2O concentration is the result of dynamic production and consumption processes in soils as soils act as sources or sinks of N2O emissions through denitrification and nitrification (Chapuis-Lardy et al., 2007). There was a slight increase of atmospheric N2O concentration during 1750-1860 according to the ice core records, but showed a rapid increase from 1860 to

present (Ciais et al., 2014). Nature sources of N2O emissions have been discussed in section 3.2 & 3.3. Previous studies found that there were some anthropogenic N2O emissions along with the natural sources in the pre-industrial era (Davidson, 2009; Syakila and Kroeze, 2011). Syakila and Kroeze (2011) found anthropogenic N2O emission began since 1500 because of the biomass burning and agriculture. The total anthropogenic N2O emission in their study was estimated as 1.1 Tg N in 1850. In addition, Davidson (2009) derived a time-course analysis of sources and sinks of atmospheric N2O since 1860. The pre-industrial anthropogenic N2O sources in his study included biomass burning, agriculture (e.g. manure and fertilizer application, and the cultivation of legume) and human sewage, the sum of which was 0.7 (0.6-0.8) Tg N yr-1 (Davidson, 2009). Thus, anthropogenic N2O emission has already existed in 1860, but in a small magnitude as compared with the contemporary amount.

Davidson (2009) mentioned that there was possibly a certain amount of N2O loss in the pre-industrial period through atmospheric sink and the reduced emission from tropical deforestation. He estimated the anthropogenic sink as 0.26 Tg N in 1860. In addition, the deforestation of tropical forest might have caused a loss of N2O emissions in 1860, which was estimated as 0.03 Tg N (Davidson, 2009). However, studies have shown that the conversion of forest to pasture and cropland could increase or have no effect on N2O emissions because the effects depended on disturbance intensity of human activities on soil conditions (van Lent et al., 2015). For instance, N2O emissions tended to increase during the first 5-10 years after conversion and thereafter might decrease to average upland forest or low canopy forest levels in the non-fertilized croplands and pastures. In contrast, emissions were at a high level during and after fertilization in fertilized croplands (van Lent et al., 2015). Thus, more work is needed to study how forest degradation affects N2O fluxes (Mertz et al., 2012).

8. Although my comments above all focus on the PI global total estimate, perhaps the more important contribution of this manuscript is the simulated spatial distribution of those PI soil emissions. It is not surprising that the model simulates the majority of

the soil emissions coming from tropical forest soils. That is also true today for nonagricultural soils. There are a few curious details that jump out at me from the map (Fig. 4). Why are emissions from the Amazon Basin and SE Asia so much lower than from the Congo Basin? Other models that I am aware of don't show that difference (e.g., Zhuang et al., 2012; Stehfest & Bouwman, 2006; Potter et al., 1996). Which of the datapoints in Fig. 3 are from tropical forests and which continents are they from? Is there validation support for the Congo having much higher emissions that the Amazon or SE Asia? More discussion would be helpful to interpret the variation shown in this map, such as where agriculture was or had been, where wetlands are, and where there are hot spots other than tropical forests. For example, I see a bunch of small red spots that appear to be near the Andes range, which puzzles me, but perhaps there is a good explanation. Ditto for why Northeastern Brazil, which is generally rather xeric, shows up as a hot spot. Also curious are the hot spots in southwestern China and the southeast coast of Australia.

(1) Why are emissions from the Amazon Basin and SE Asia so much lower than from the Congo Basin? Other models that I am aware of don't show that difference (e.g., Zhuang et al., 2012; Stehfest & Bouwman, 2006; Potter et al., 1996).

Response: There are three major explanations for the spatial pattern differences among various studies. Firstly, the vegetation map in our study includes at most five biome types (at most four natural vegetation types and one crop type) within each grid cell. For example, in the Congo and Amazon Basin, the major natural vegetation type is Tropical Broadleaf Evergreen Forest (TrBEF). Many other models only include one vegetation type within each grid cell. This difference can cause large difference in spatial distribution of N2O emissions between our results and other model simulations.

Secondly, DLEM simulates both soil nitrogen transformation and nitrogen export or leaching into riverine ecosystems (see section 2.1). Many other models don't simulate nitrogen leaching or export. In our simulation, we found that high rainfall (especially heavy rainfall events) can cause a large amount of available nitrogen exports to riverine

ecosystems and thus reduce soil available N and N2O emissions in these grid cells.

The third cause may be the difference in model driving data. Stehfest & Bouwman (2006) used the mean annual precipitation and annual temperature developed by New et al. (1999) during 1961-1990. In Zhuang et al., (2012), they used the monthly data from the original literature and a historical climate database from the Climate Research Unit during 1961-2002 (Mitchell and Jones, 2005). While, DLEM used the long-term mean climate datasets (daily CRUNCEP climate data) from 1901-1930 to represent the initial climate state in 1860.

In the two studies mentioned above, they stated that soil and climate characteristics are major factors that affect N2O emissions. Unfortunately, neither of them showed the spatial distribution of precipitation or temperature, and the correlation between the climate and N2O emissions. Moreover, through comparing the spatial N2O emission map from those studies, we found that the distributions and magnitudes of emissions in the Congo, Amazon Basin, and Southeast Asia also differed significantly. The spatial patterns of annual precipitation and temperature in this study are shown in Fig. S2. The Congo, Amazon Basin, and Southeast Asia are located in the tropics. The three regions have similar annual temperature (Fig. S2a), but have significantly different annual precipitation (Fig. S2b). In some areas in Amazon Basin and Southeast Asia, the annual precipitation was even higher than 3000 mm. In contrast, the annual precipitation in the Congo varied from 1300 to 2000 mm. In the DLEM, we explicitly considered the daily N leaching and runoff. Because of the heavy rainfall in the Amazon Basin and Southeast Asia, more N might be leached from the soil during the wet season, which could cause the lower annual N2O emissions. In addition, both denitrification and nitrification are highly affected by the soil water content. As field experiments revealed N2O or NO could be reduced into N2 when soils are in saturation (Davidson et al., 2000), DLEM also represent the formation and proportion of N2O in total nitrogen oxides, considering the effect of soil moisture change. Thus, excessive soil water content during the wet season in Amazon Basin and Southeast Asia might reduce the activities

of microbes, thus causing smaller amount of N2O emission.

(2) Which of the datapoints in Fig. 3 are from tropical forests and which continents are they from?

Response: Firstly, we are sorry for the mistake in this Figure. The x-axis should be "observed N2O emission" rather than "simulated N2O emission". In the new version, we redraw this figure. In addition, we used different symbols to mark all sites in Fig. 3 to make it clearly show the locations of all 20 sites. The information of each site can be found in the supplementary material (Table S1).

(3) Is there validation support for the Congo having much higher emissions that the Amazon or SE Asia?

Response: There were only two sites in the validation from southeast Asia (site 14) and the east coast of Austria (site 10). Unfortunately, there was no available validation to support the arguments that the Congo has much higher emissions than the Amazon Basin or Southeast Asia. However, we did find some measurements in Kim et al. (2016), which could support our estimates in the central Africa. In their study, they calculated the average N2O emission from ten observations in the Congo Basin, which was 4.2±1.5 kg N ha-1 yr-1 and close to our estimates in this region.

(4) More discussion would be helpful to interpret the variation shown in this map, such as where agriculture was or had been, where wetlands are, and where there are hot spots other than tropical forests.

Response: The spatial distributions of cropland and wetlands have been provided in the manuscript. Meanwhile, the emission from pre-industrial cropland was discussed in line 227-236. It is hardly to make sure the certain crop types 150 years ago. Thus, N2O emission from cropland remained quite uncertain. For the N2O emission from wetlands and peatlands, we have discussed in line 248-258. We did not include the estimate of pre-industrial wetlands or peatlands because of the uncertainty of wetland

area and distribution, but it will be included in the future study. The results in the DLEM simulation indicated that where natural vegetation was, specifically the tropical forest, were hot spots for N2O emissions in the pre-industrial period. Some scattered hot spots the reviewer mentioned were also from the tropics as described below.

(5) I see a bunch of small red spots that appear to be near the Andes range, which puzzles me, but perhaps there is a good explanation. Ditto for why Northeastern Brazil, which is generally rather xeric, shows up as a hot spot. Also curious are the hot spots in southwestern China and the southeast coast of Australia.

Response: We have noticed those "hot spots". Near the Andes range and in south-western China, those mountains have higher altitudes and smaller amount of annual precipitation compared with the adjacent basins. Less N leaching happened in those regions. Meanwhile, both regions in the tropics that are dominant with TrBEF. Thus, it is possible that N2O emission was higher in this circumstance. In the Northeastern Brazil and the Southeast coast of Australia, both regions are along the coast. Both regions were not xeric according to the annual precipitation data used in this study. In the Northeastern Brazil, the dominant vegetation type is still TrBEF. Similarly, less N leaching and proper soil water content might cause higher amount of N2O emissions. In the east coast of Australia, anthropogenic activities contributed a large amount of N deposition in 1860 compared to other regions of Australia. Several grids with higher emissions were dominant with Temperate Broadleaf Evergreen Forest (TBEF). Mean-while, the precipitation was generally higher along the Australian coast. Thus, higher N deposition with proper precipitation might cause this high N2O emission.

Technical Points 1. Line 41: This statement ignores that some anthropogenic emissions were already present prior to or at the beginning of the industrial revolution.

Response: It is true that there existed anthropogenic N2O emission before 1860; however, the total amount is substantially lower than the contemporary human-induced N2O emissions. The description of anthropogenic emissions in the pre-industrial era

was added, shown as "Human-induced biogenic N2O emissions are calculated by sub-tracting the pre-industrial emissions (Tian et al., 2016), even though a small amount of anthropogenic N2O emissions was present before 1860, which was estimated as 1.1 Tg N yr-1 in 1850 by Syakila and Kroeze (2011) and 0.7 (0.6-0.8) Tg N yr-1 in 1860 by Davidson (2009)."

2. Line 55: Add recent results from Prather et al. 2015.

Response: The latest study done by Prather et al. (2015) has been added into line 56, shown as "Prather et al. (2015) provided an estimate of the pre-industrial emissions (total natural emission: 10.5 Tg N yr-1) based on the then-most-recent model study with a corrected lifetime of $116\pm9$ years."

3. Line 70: Change "is" to "are" because the word "data" is plural: "the data are". Response: It has been revised.

4. Line 178: Use estimate from Prather et al. 2015.

Response: We have removed the section 2.4.2 and section 3.2, which described the one-box model validation of simulation results. Thus, there is no need to replace the N2O lifetime in this section. In addition, we added the comparison of the estimate in this study with the estimation by Prather et al. (2012, 2015) in the section 3.2.

5. Line 312: Consider other estimates, such as those of Voss et al. 2013.

Response: We carefully read the paper from Voss et al. (2013) and found that their estimates were directly from the IPCC AR4 (Table 7.7 in Chapter 7). In their paper, the N2O emission from ocean was 5.5 Tg N yr-1 because they considered the emissions (1.7 Tg N yr-1) from "rivers, estuaries, and coastal zones" as the marine emissions. Thus, the average marine emissions are 3.8 Tg N yr-1, as shown in Table 6.9 of Chapter 6 in the IPCC AR5.

6. Figure 2. I don't understand the units. How can these units of crop area apply to each individual pixel?

Response: To avoid the confusion of the unit, it has been changed from "km2" to "km2/grid". The size of individual pixel is 0.5 degree, equivalent to around 2500 km2 at the equator. Meanwhile, we have crop area fraction in each pixel (mentioned in the section 2.1 & 2.2). Then, in each grid, crop area fraction multiplying the pixel size represents the crop area. The numbers in the legend mean the cropland area in each 0.5-degree pixel.

7. Figure 3. The data used for this graph should be referenced.

Response: All papers that used for the graph cited in the new version.

8. Figure 5. The bottom panel is all that is needed. The top panel is redundant. However, you could also add a panel of mean flux per hectare, which would be useful, because it is difficult to compare fluxes across continents when the contents have such different total areas.

Response: We agree with the reviewer. We have removed the top panel. Instead, we added a panel of N2O emission rates per unit area (g N m-2 yr-1) with uncertainty ranges at continental-level in 1860, as shown in Fig. 5 (a).

9. Figure 6. The two panels are largely redundant. The pie chart could include both the percentage of the total and the estimate of Tg/yr, which would obviate the need for the upper panel. However, again, the mean flux per hectare by biome would be an interesting panel to add.

Response: We agree with the reviewer. We have removed the top panel. Instead, we added a panel of N2O emission rates per unit area (g N m-2 yr-1) with uncertainty ranges at biome-scale in 1860, as shown in Fig. 6 (a). In addition, we added the biome-scale emission amounts and their uncertainty ranges into the pie chart, as shown in Fig. 6 (b).

10. Table 2. The number of significant figures shown is excessive. I suggest rounding to the nearest Gg. The uncertainties are such that any fraction of a Gg is meaningless.

[Figure]

Response: Since the one-box model section has been removed, Table 1 was deleted, and "Table 2" was changed to "Table 1". The uncertainties have been removed. We added the biome- and continental-scale N2O emissions in the supplementary material (Table S2). For the mean annual N2O emissions (Tg N yr-1) and emission rate per unit area (kg N ha-1 yr-1), we have listed all numbers in the Table S3. We included the revised figures as below:

References: Chapuis‐Lardy, L., Wrage, N., Metay, A., CHOTTE, J. L., and Bernoux, M.: Soils, a sink for N2O? A review, Global Change Biology, 13, 1-17, 2007.

Ciais, P., Sabine, C., Bala, G., Bopp, L., Brovkin, V., Canadell, J., Chhabra, A., DeFries, R., Galloway, J., Heimann, M. and Jones, C.: Carbon and other biogeochemical cycles. In Climate Change 2013: The Physical Science Basis. Contribution of Working Group I to the Fifth Assessment Report of the Intergovernmental Panel on Climate Change. Cambridge University Press, 465-570, 2014.

Davidson, E. A., Keller, M., Erickson, H. E., Verchot, L. V., and Veldkamp, E.: Testing a Conceptual Model of Soil Emissions of Nitrous and Nitric Oxides: Using two functions based on soil nitrogen availability and soil water content, the hole-in-the-pipe model characterizes a large fraction of the observed variation of nitric oxide and nitrous oxide emissions from soils, Bioscience, 50, 667-680, 2000.

Davidson, E. A.: The contribution of manure and fertilizer nitrogen to atmospheric nitrous oxide since 1860, Nature Geoscience, 2, 659-662, 2009.

Davidson, E. A., and Kanter, D.: Inventories and scenarios of nitrous oxide emissions, Environmental Research Letters, 9, 105012, 2014.

Denman K., Brasseur G., Chidthaisong A., Ciais P. M., Cox P., Dickinson R., Hauglustaine D., Heinze C., Holland E., Jacob D., Lohmann U., Ramachandran S., da Silva Dias P., Wofsy S. and Zhang X.: Couplings Between Changes in the Climate System and Biogeochemistry. In: Climate Change 2007: The Physical Science Basis. Contri-

bution of Working Group I to the Fourth Assessment Report of the Intergovernmental Panel on Climate Change [Solomon S., Qin D., Manning M., Chen Z., Marquis M., Averyt K. B., Tignor M., and Miller H. (eds.)]. Cambridge University Press, Cambridge, United Kingdom and New York, NY, USA, 501-566, 2007.

Mertz, O., Müller, D., Sikor, T., Hett, C., Heinimann, A., Castella, J.-C., Lestrelin, G., Ryan, C. M., Reay, D. S., Schmidt-Vogt, D., Danielsen, F., Theilade, I., Noordwijk, M. v., Verchot, L. V., Burgess, N. D., Berry, N. J., Pham, T. T., Messerli, P., Xu, J., Fensholt, R., Hostert, P., Pflugmacher, D., Bruun, T. B., Neergaard, A. d., Dons, K., Dewi, S., Rutishauser, E., Sun, and Zhanli: The forgotten D: challenges of addressing forest degradation in complex mosaic landscapes under REDD+, Geografisk Tidsskrift-Danish Journal of Geography, 112, 63-76, 10.1080/00167223.2012.709678, 2012.

Mitchell, T. D., and Jones, P. D.: An improved method of constructing a database of monthly climate observations and associated high‐resolution grids, International journal of climatology, 25, 693-712, 2005.

New, M., Hulme, M., and Jones, P.: Representing twentieth-century space–time climate variability. Part I: Development of a 1961–90 mean monthly terrestrial climatology, Journal of climate, 12, 829-856, 1999.

Prather, M. J., Holmes, C. D., and Hsu, J.: Reactive greenhouse gas scenarios: Systematic exploration of uncertainties and the role of atmospheric chemistry, Geophysical Research Letters, 39, 2012.

Prather, M. J., Hsu, J., DeLuca, N. M., Jackman, C. H., Oman, L. D., Douglass, A. R., Fleming, E. L., Strahan, S. E., Steenrod, S. D., and Søvde, O. A.: Measuring and modeling the lifetime of nitrous oxide including its variability, Journal of Geophysical Research: Atmospheres, 120, 5693-5705, 2015.

Stehfest, E., and Bouwman, L.: N2O and NO emission from agricultural fields and soils under natural vegetation: summarizing available measurement data and modeling of

global annual emissions, Nutrient Cycling in Agroecosystems, 74, 207-228, 2006.

Syakila, A., and Kroeze, C.: The global nitrous oxide budget revisited, Greenhouse Gas Measurement and Management, 1, 17-26, 2011.

Kim D. G., Thomas, A. D., Pelster D., Rosenstock, T. S., Sanz-Cobena A.: Greenhouse gas emissions from natural ecosystems and agricultural lands in sub-Saharan Africa: synthesis of available data and suggestions for further research, Biogeosciences, 13, 4789, 2016.

Tian, H., Lu, C., Ciais, P., Michalak, A. M., Canadell, J. G., Saikawa, E., Huntzinger, D. N., Gurney, K. R., Sitch, S., and Zhang, B.: The terrestrial biosphere as a net source of greenhouse gases to the atmosphere, Nature, 531, 225-228, 2016.

van Lent, J., Hergoualc'h, K., and Verchot, L.: Reviews and syntheses: Soil N2O and NO emissions from land use and land use change in the tropics and subtropics: a meta-analysis, Biogeosciences, 12, 2015.

Voss, M., Bange, H. W., Dippner, J. W., Middelburg, J. J., Montoya, J. P., and Ward, B.: The marine nitrogen cycle: recent discoveries, uncertainties and the potential relevance of climate change, Philosophical Transactions of the Royal Society B: Biological Sciences, 368, 20130121, 2013.

Zhuang, Q., Lu, Y., and Chen, M.: An inventory of global N2O emissions from the soils of natural terrestrial ecosystems, Atmospheric Environment, 47, 66-75, 2012.

Please also note the supplement to this comment:
http://www.clim-past-discuss.net/cp-2016-103/cp-2016-103-AC1-supplement.pdf
* * *
[Figure]

Fig. 2 The spatial distribution of cropland area in 1860.

**Fig. 1.**

[Figure]

**Fig. 3** The comparison of the DLEM-simulated N$_2$O emissions with field observations. All sites were described in the supplementary material (Table S1).

**Fig. 2.**

[Figure]

**Fig. 5** Estimated $N_2O$ emission rates (a) and emissions (b) with uncertainty ranges at continental-level in 1860.

Solid line within each box refers to the median value of $N_2O$ emission rate or amount.

**Fig. 3.**

[Figure]

Fig. 6 (a) Estimated N$_2$O emission rate at biome-level in 1860 with the median value (solid line), the mean (solid dot), and the uncertainty range of emission rates from different biomes. The emission rate in the tundra was removed because of the extremely small value (less than 0.003g N m$^{-2}$ yr$^{-1}$); (b) Estimated N$_2$O emission (Tg N yr$^{-1}$) with uncertainty ranges and its percentage (%) at biome-level in 1860.

**Fig. 4.**

[Figure]

Fig. S2 (a) The average annual temperature during 1901–1930; (b) The average annual precipitation during 1901–1930.

**Fig. 5.**

---

## Author Comment (AC2) · 26 Mar 2017

Dear reviewer #2,

Many thanks for your highly valuable comments! All your questions have been answered as follows:

1. The country-level analysis does not make much sense as a large amount of countries had different boundaries compared to present. In line 396, those country-level emissions might need to be removed.

Response: Yes, the current country boundaries are different from that in the preindustrial era. Here we just want to look at the regional differences in N2O emission for current country-level from geographical perspective. The region division based on

country scale could be more interesting, so we still hope to keep this country-level analysis here.

2. I am little curious to see the small uncertainties in continent-level N2O show in Figure 5 as the LHS was used and the large uncertainties were shown in below panel in Figure 5.

Response: The small uncertainty range shown in the upper panel of Fig. 5 was the 95% confidence interval of the mean estimate, as explained in the manuscript. The uncertainty range of pre-industrial N2O emissions was present using the minimum and maximum estimate (4.76-8.13 Tg N yr-1) in this study, which was consistent with other studies, such as the reported estimates in the IPCC AR5. Here, the Bootstrap resampling method was used to define the uncertainty bounds of global mean N2O emission (6.20 Tg N yr-1) (shown in line 216-219 of previous manuscript). It was used to verify the stability of the LHS approach. The 95% confidence intervals (6.03-6.36 Tg N yr-1) of the mean did not represent the uncertainty range for pre-industrial N2O emission in this study. In order to avoid the confusion, we will not report this narrow range in the revised manuscript.

Meanwhile, the first reviewer also suggested to remove it because the upper and below panel deliver the same information. Instead, we replaced the Fig. 5(a) with a panel of N2O emission rates per unit area (g N m-2 yr-1) with uncertainties.

3. The model implementation is not clear. I assume this study is based on a steady state or semi-steady state simulation. The equilibrium run was for 1860, followed by a spinup. The transient run was driven with climate data in 1860 (line 153). What is the data source? If the equilibrium run was based on 1860 data (most). Then, there are small discrepancies among spinup and transient runs. A comparison between equilibrium and transient run might be needed. If there are no big differences, using equilibrium run might be more convincing, as most driving forces were 1860 except climate data of 1901-1930. If the authors really want to have a transient run, the

model simulations should start even further to capture the legacy impacts of natural and anthropogenic impacts, particularly the land use change.

Response: Yes, this study was based on steady state simulation. The data sources for equilibrium run were all based on the data in 1860. Our transient run for 1860 was actually an extension of the equilibrium run. We don't have transient data before 1860 to realistically include the legacy effects from land use change, climate, etc. before 1860. The reason we ran this transient run was to avoid the abnormal fluctuations after equilibrium run, rather than capturing the legacy impacts. Fig. 4 in the manuscript is the result from equilibrium run. We made a comparison between the equilibrium and transient results for 1860 (Fig. S3). Although there were small differences for some grid cells between the two simulation results, the simulation results for the equilibrium run were similar to the transient run as a whole.
* * *
[Figure]

Fig. S3 (a) The spatial distribution of global $N_2O$ emission from the equilibrium run; (b) The spatial pattern distribution of global $N_2O$ emission from the transient run.

**Fig. 1.**

---

## Author Response (AR1)

Dear reviewer #1,

Thanks very much for the precious comments and suggestions. All the comments and suggestions are addressed in the revised manuscript.

1. With respect to constraints on the overall PI global emissions of $N_2O$, I have more confidence in the top-down approach using atmospheric concentrations and lifetimes of $N_2O$, than the bottom up simulations of a highly parameterized process model. The most recent top-down estimate (Prather et al., 2015) is cited in passing by the authors, but the estimates are not included in the present manuscript. The estimates from the IPCC AR4 and from Davidson & Kanter (2014), mentioned in lines 53-54, were based largely on the 2012 paper by Prather et al., but their 2015 paper provides an important update on lifetime estimates and resulting PI emission estimates. They now recommend using lifetimes of 123 years for PI and 116 years for the present (+/- 9 years), and from those lifetime estimates, they derive a new PI emission estimate of 10.5 Tg/yr. Fortunately, this is very close to other estimates, including the one from this study. Nevertheless, it should be specifically cited.

Response:

We have cited the recent study by Prather et al. (2015) in the introduction and discussion sections of the revised manuscript.

Line 55-57: Prather et al. (2015) provided an estimate of the pre-industrial emissions (total natural emission: 10.5 Tg N $yr^{-1}$) based on the most recent study with a corrected lifetime of 116 years.

Line 267-275: "Top-down" methodology used to estimate $N_2O$ emissions is based on atmospheric measurements and an inversion model (Thompson et al. 2014). Prather et al. (2012) provided an estimate of 9.1±1.0 Tg N $yr^{-1}$ of natural emission in the pre-industrial era using observed pre-industrial abundances of 270 ppb and model estimates of lifetime decreased from 142 years in the pre-industrial era to 131±10 years in the present-day. Later, Prather et al. (2015) re-evaluated $N_2O$ lifetime based on Microwave Limb Sounder satellite measurements of stratospheric, which was consistent with modeled values in the present-day. The lifetime in the pre-industrial era and present-day was estimated to be 123 and 116±9 years, respectively. The current lifetime increases the pre-industrial natural emission from 9.1±1.0 to 10.5 Tg N $yr^{-1}$.

2. The point that the lifetime has probably decreased since PI times should be discussed. As far as I can tell, a varying lifetime cannot be incorporated into the one-box model (line 171) used by the authors. Perhaps the resulting global estimate is not terribly sensitive to this change, but that should be evaluated and discussed.

Response:

In the revised version, we have removed the one-box model validation. In addition, we added the discussion of the decreased lifetime since PI times, which was also mentioned in the response to question #1 (Line 267-275).

3. I fail to see how the analysis presented in Figure 7 and Table 1 provides additional confidence in the summed global estimate from this study. I can see the value of a sensitivity analysis of initial PI atmospheric concentrations and lifetimes, which Prather's papers have already done and for which they could be cited. In contrast, the analysis in Fig. 7 and Table 1 is clouded by the unclear source of annual emissions over the simulated time period and the validity of those assumptions. The text (lines 182- 185) suggests that model output was used for annual emission estimates: "The mean with 95% confidence intervals, the maximum, and minimum values of estimates from DLEM simulations were applied as initial emissions to calculate the atmospheric $N_2O$ concentration in 2006 as shown in Table 1 (Scenarios 1–4 and baseline), as well as concentration changes from 1860 to 2006, as shown in Figure 7." However, the Fig. 7 captions indicates that the "net additions of anthropogenic $N_2O$ emission amount in different years were listed in Syakila and Kroeze, 2011." I don't understand which was used to estimate annual increments of $N_2O$ concentration in Fig 7 – was it model output, as indicated on lines 182-185, or was it the net additions estimated by S&K as indicated in the figure caption? Both have problems. S&K estimated fairly substantial $N_2O$ emissions from agriculture during the late 19th and early 20th centuries, but they also estimated a rather large decrease in natural emissions compared to 1500 (which are very difficult to estimate, see my further comments below), so their estimate of the net change relative to 1500 was small for this time period. However, the starting point for the present study is 1860. Therefore, it is incorrect to subtract this decline in natural emissions that preceded 1860 from the growth in anthropogenic emissions since 1860. S&K did this to show changes since their starting point of 1500, but using their "net additions" column without accounting for a different starting point in the present study introduces a significant bias. It is the net change relative to 1860 that is important for the present study, so the "net additions" estimated by S&K should be recalculated relative to 1860 if they are to be used in the analysis for Table 1 and Fig. 7.

I showed in my 2009 paper, and Smith et al. (2012) have affirmed, that atmospheric $N_2O$ began rising significantly many decades before fertilizer use became common in the 1950s, and so the "net additions" to the atmosphere must have been larger than those estimated by S&K relative to 1500, although they may be similar if they were corrected to be relative to 1860. We speculate that this increase in emissions between 1860 and 1950 was due to mineralization of soil N as agriculture expanded into regions of previously untilled soils, thus mobilizing N for rapid cycling, including a fraction lost at $N_2O$. I also suspect that the current DLEM may not include effects of soil mining when virgin soil is first tilled, so if Table 1 is based on DLEM simulations, as indicated in the text on lines 182-185, then I suspect emissions from 1860 to 1950 were underestimated, which would affect the slope of the trend line later in the analysis as well.

I realize that the point of Figure 7 is not the accuracy of the simulated trend line, but rather the end point, but if the trend line agrees so poorly with the observations, then one has to question the validity of the model and the input data, which calls into question the reliability of the end point analysis. I believe that Fig. 7 and Table 1 could be replaced with citations of the sensitivity analyses done by Prather et al. (2012, 2015), but if the authors persist in wanting to include their own analysis, I would suggest that they utilize another source of "net addition" emissions than those of S&K relative to 1500.

Response:

According to your suggestions, we have removed the one-box model validation. Instead, we cited the work done by Prather et al. (2012, 2015) and compared our results with theirs in the section 3.2.

Line 276−287: Natural sources for $N_2O$ include soil under natural vegetation, oceans, and atmospheric chemistry (Ciais et al., 2014). The emission from atmospheric chemistry was estimated as 0.6 with an uncertainty range of 0.3−1.2 Tg N $yr^{-1}$. Syakila and Kroeze (2011) estimated global natural emissions from oceans as 3.5 Tg N $yr^{-1}$. Oceanic emission was estimated as 3.8 with an uncertainty range of 1.8−5.8 Tg N $yr^{-1}$ in the IPCC AR4. However, the uncertainty range became larger (1.8−9.4 Tg N $yr^{-1}$) in the IPCC AR5. In our study, the simulated $N_2O$

emission was from agricultural and natural soils. The natural emission was estimated as 5.78 (4.4–7.72) Tg N yr$^{-1}$. Combining the atmospheric chemistry and the ocean emissions in the IPCC AR5 with the natural emissions from our study, the global total natural $N_2O$ emissions were 10.18 (6.5-18.32) Tg N yr$^{-1}$. The large uncertainty range was attributed to the uncertainty from oceanic emission, atmospheric chemistry emission, and our estimation. The estimated global total amount (10.18 Tg N yr$^{-1}$) in this study was comparable to the estimate (10.5 Tg N yr$^{-1}$) by Prather et al. (2015) using the top-down approach.

4. The change in "natural" emissions before and after 1860 should be discussed. As I noted above, S&K deduce a substantial decline in natural emissions from 1500 to 1850. Similarly, I included a significant change in non-agricultural soil emissions due to tropical deforestation, which began growing rapidly in the late 20th century (Davidson 2009). Whether pre-1850 or post-1950, these changes in natural soil emissions are difficult to estimate, but the uncertainties that they represent should be considered, and biases resulting from how they are or are not included should be considered.

Response:

We agree that different factors caused different variation patterns in $N_2O$ fluxes before and after 1860. We did not consider the pre-1850 natural emission change because we assumed emission in 1860 can represent the pre-industrial level although it has declined from 1500 to 1850. Our estimation from the process-based model can capture the $N_2O$ emission due to land use change in the late 20$^{th}$ century, but it is beyond the scope of this paper. Since pre-industrial $N_2O$ emission is not always stable and remains a large uncertainty, our estimation can only go back to 1860 and represent $N_2O$ level before intensive human disturbance.

5. While the top-down approach of Prather et al. (2012, 2015) and the one box model used in the present study help constrain total PI emissions, the soil emission estimate must still be made by difference between total emissions and oceanic emissions. While the AR5 estimate of 3.8 Tg $N_2O$-N/yr (range: 1.8 - 9.4; Ciais et al., 2013) is widely cited for emissions from the oceans, it is highly uncertain, so simply subtracting 3.8 (or 3.5 – 4.5 as in Table 1 of the present manuscript) from a total PI source estimate of about 11 Tg $N_2O$-N/yr (+/-1) doesn't really narrow the confidence estimate of the PI terrestrial source a great deal. Indeed, I just discovered a curious inconsistency between the AR5 best estimate of 3.8 with a review paper by Voss et al. (2013), which cites that same 3.8 value for $N_2O$ emissions from the open ocean, but then adds another 1.7 Tg $N_2O$-N/yr for emissions from the continental shelf regions. I don't know if the AR5 review of the literature failed to adequately represent continental shelf regions or if Voss et al. might be double accounting. If Voss et al. are correct, the AR5 estimate of oceanic emissions may be biased toward the low end, which would mean that the terrestrial PI source may more likely be in the range of 5 Tg $N_2O$-N/yr or less. In any case, this highlights how uncertain the oceanic estimate is, which means we have to have similar uncertainty in the estimate of the PI terrestrial source. The narrow range of uncertainty in the present study's PI terrestrial source (6.03−6.36 Tg $N_2O$-N/yr) reported on line 331 is unrealistically small.

Response:

Yes, the soil emission estimation must still be made by difference between total emissions and oceanic emissions regardless of methodology (top-down or bottom-up). In the IPCC AR5, the average oceanic emission is 3.8 Tg N $yr^{-1}$, with a larger uncertainty range compared with the estimate in the AR4. The estimate from Voss et al. (2013) indicated that oceanic emission was 1.7 Tg N $yr^{-1}$ more than the average in the AR5. It is because they considered the emissions (1.7 Tg N $yr^{-1}$) from "rivers, estuaries, and coastal zones" as the marine emissions, as written in Table 7.7 of the IPCC AR4 Chapter 7. Thus, the average estimation in AR5 is still trustable. In this study, to compare with the results (10.5 Tg N $yr^{-1}$) in Prather et al. (2015), we need to sum our estimate and other natural emissions. The global total natural $N_2O$ emissions were 10.18 (6.5-18.32) Tg N $yr^{-1}$ in the preindustrial era.

The small uncertainty range shown in the upper panel of Fig. 5 was the 95% confidence interval of the mean estimate, as explained in the manuscript. The uncertainty range of pre-industrial $N_2O$ emissions was present using the minimum and maximum estimate (4.76−8.13 Tg N $yr^{-1}$) in this study, which was consistent with other studies, such as the reported estimates in the IPCC AR5. Here, the Bootstrap resampling method was used to define the uncertainty bounds of global mean $N_2O$ emission (6.20 Tg N $yr^{-1}$) (shown in line 216-219 of previous manuscript). It was used to verify the stability of the LHS approach. The 95% confidence intervals (6.03-6.36 Tg N $yr^{-1}$) of the mean did not represent the uncertainty range for pre-industrial $N_2O$ emission in this study. Thus, we will not report this narrow range in the revised manuscript to avoid the confusion.

6. The authors have misunderstood the emission estimates from my 2009 paper, which they incorrectly describe on lines 299-301: "However, the indirect emissions from the riverine induced by the leaching and runoff of manure applications in agro-ecosystems, legume crop N fixation, and human sewage discharging have not been addressed in Davidson (2009)." On the contrary, I derived emissions factors from a statistical model that was constrained by the historical record of atmospheric concentrations and fertilizer and manure use, so the emission factors derived from that analysis necessarily included all of the emissions, direct and indirect, that could be statistically correlated with historical fertilizer and manure use ("The sources attributed to fertilizers and manures include indirect emissions from downwind and downstream ecosystems, including human sewage." Davidson, 2009). Therefore, it is incorrect for the authors to calculate an additional indirect source (line 305) using IPCC default factors to add onto the estimate that they took from my paper that they misunderstood to be only direct emissions. They could either use an unmodified estimate from my paper or they could derive a new one, based on IPCC default values for both direct and indirect emissions based on estimates of BNF, fertilizer-N, and manure-N for 1860. Furthermore, note that the 0.42 Tg $N_2O$-N/yr that they extracted from my paper for 1860 was for anthropogenic biological emissions (i.e., soils) only, and that there were also some other anthropogenic emissions at that time, such as biomass burning (see SI for Davidson 2009).

Response:

We are sorry for the misunderstanding of this paper. We deleted this sentence and recalculated the overall $N_2O$ emissions from this paper. In Davidson (2009), two approaches (top-down and bottom-up) had been applied to estimate the anthropogenic biogenic $N_2O$ emissions in 1860. The estimates from top-down and bottom-up were 0.42 and 0.54 Tg N yr$^{-1}$, respectively. Thus, the final number we used in this study was 0.5 with an uncertainty range of 0.4–0.6 Tg N yr$^{-1}$. In addition, $N_2O$ from the biomass burning was assumed to be 0.2 Tg N yr$^{-1}$ in 1860 in Davidson (2009). In sum, the total anthropogenic $N_2O$ emission in 1860 was estimated as 0.7 (0.6–0.8) Tg N yr$^{-1}$ in Davidson (2009). We added below content in the revised version:

Line 332-334: The pre-industrial anthropogenic $N_2O$ sources in his study included biomass burning, agriculture activities (e.g., manure application, and the cultivation of legume) and human sewage, the sum of which was 0.7(0.6–0.8) Tg N yr$^{-1}$ (Davidson, 2009).

7. The authors should also acknowledge that there were anthropogenic effects on the $N_2O$ budget before 1860, so the 1860 fluxes don't necessarily represent only "natural" emissions. This includes some $N_2O$ from agricultural expansion that mined soil N and also added BNF, some biomass burning, a tiny amount of industrial and transportation sector emissions, and possibly a loss of emissions from degraded natural soils that had been plowed for centuries or millennia, some of which were highly eroded.

Response:

Yes, the 1860 fluxes don't necessarily represent only "natural" emissions. In our study, when we mentioned "natural" emissions, we excluded the emissions from cropland soils. Our study has only addressed the anthropogenic emissions from cropland expansion and manure application, but we are unable to simulate the anthropogenic emissions from biomass burning and other sectors. As described in the response 6, we have added the discussion on the pre-industrial anthropogenic $N_2O$ emission in this manuscript (Section 3.4 Line 332-334).

See the section 3.4, Line 322-346:

**3.4 The $N_2O$ budget in the pre-industrial era**

The observed $N_2O$ concentration reflects the result of dynamic production and consumption processes in soils as soils act as sources or sinks of $N_2O$ emissions through denitrification and nitrification (Chapuis-Lardy et al., 2007). There was a slight increase of atmospheric $N_2O$ concentration during 1750−1860 according to the ice core records, but showed a rapid increase from 1860 to present (Ciais et al., 2014). Nature sources of $N_2O$ emissions have been discussed in section 3.2 & 3.3. Previous studies found that there were some anthropogenic $N_2O$ emissions along with the natural sources in the pre-industrial era (Davidson, 2009; Syakila and Kroeze, 2011). Syakila and Kroeze (2011) found anthropogenic $N_2O$ emission began since 1500 because of the biomass burning and agriculture. The total anthropogenic $N_2O$ emission in their study was estimated as 1.1 Tg N in 1850. In addition, Davidson (2009) derived a time-course analysis of sources and sinks of atmospheric $N_2O$ since 1860. The pre-industrial anthropogenic $N_2O$ sources in his study included biomass burning, agriculture (e.g. manure application, and the cultivation of legume) and human sewage, the sum of which was 0.7 (0.6−0.8) Tg N yr$^{-1}$ (Davidson, 2009). Thus, anthropogenic $N_2O$ emission has already existed in 1860, but in a small magnitude as compared with the contemporary amount.

Davidson (2009) mentioned that there was possibly a certain amount of $N_2O$ loss in the pre-industrial period through atmospheric sink and the reduced emission from tropical deforestation. He estimated the anthropogenic sink as 0.26 Tg N in 1860. In addition, the deforestation of tropical forest might have caused a loss of $N_2O$ emissions in 1860, which was estimated as 0.03 Tg N (Davidson, 2009). However, studies have shown that the conversion of forest to pasture and cropland could increase or have no effect on $N_2O$ emissions because the effects depended on disturbance intensity of human activities on soil conditions (van Lent et al., 2015). For instance, $N_2O$ emissions tended to increase during the first 5−10 years after conversion and thereafter might decrease to average upland forest or low canopy forest levels in the non-fertilized croplands and pastures. In contrast, emissions were at a high level during and after fertilization in fertilized croplands (van Lent et al., 2015). Thus, more work is needed to study how forest degradation affects $N_2O$ fluxes (Mertz et al., 2012).

8. Although my comments above all focus on the PI global total estimate, perhaps the more important contribution of this manuscript is the simulated spatial distribution of those PI soil emissions. It is not surprising that the model simulates the majority of the soil emissions coming from tropical forest soils. That is also true today for nonagricultural soils. There are a few curious details that jump out at me from the map (Fig. 4). Why are emissions from the Amazon Basin and SE Asia so much lower than from the Congo Basin? Other models that I am aware of don't show that difference (e.g., Zhuang et al., 2012; Stehfest & Bouwman, 2006; Potter et al., 1996). Which of the datapoints in Fig. 3 are from tropical forests and which continents are they from? Is there validation support for the Congo having much higher emissions that the Amazon or SE Asia? More discussion would be helpful to interpret the variation shown in this map, such as where agriculture was or had been, where wetlands are, and where there are hot spots other than tropical forests. For example, I see a bunch of small red spots that appear to be near the Andes range, which puzzles me, but perhaps there is a good explanation. Ditto for why Northeastern Brazil, which is generally rather xeric, shows up as a hot spot. Also curious are the hot spots in southwestern China and the southeast coast of Australia.

(1) Why are emissions from the Amazon Basin and SE Asia so much lower than from the Congo Basin? Other models that I am aware of don't show that difference (e.g., Zhuang et al., 2012; Stehfest & Bouwman, 2006; Potter et al., 1996).

Response:

There are three major explanations for the spatial pattern differences among various studies. Firstly, the vegetation map in our study includes at most five biome types (at most four natural vegetation types and one crop type) within each grid cell. For example, in the Congo and Amazon Basin, the major natural vegetation type is Tropical Broadleaf Evergreen Forest (TrBEF). Many other models only include one vegetation type within each grid cell. This difference can cause large difference in spatial distribution of $N_2O$ emissions between our results and other model simulations.

Secondly, DLEM simulates both soil nitrogen transformation and nitrogen export or leaching into riverine ecosystems (see section 2.1). Many other models don't simulate nitrogen leaching or export. In our simulation, we found that high rainfall (especially heavy rainfall events) can cause a large amount of available nitrogen exports to riverine ecosystems and thus reduce soil available N and $N_2O$ emissions in these grid cells.

The third cause may be the difference in model driving data. Stehfest & Bouwman (2006) used the mean annual precipitation and annual temperature developed by New et al. (1999) during 1961−1990. In Zhuang et al., (2012), they used the monthly data from the original literature and a historical climate database from the Climate Research Unit during 1961−2002 (Mitchell and Jones, 2005). While, DLEM used the long-term mean climate datasets (daily CRUNCEP climate data) from 1901−1930 to represent the initial climate state in 1860.

In the two studies mentioned above, they stated that soil and climate characteristics are major factors that affect $N_2O$ emissions. Unfortunately, neither of them showed the spatial distribution of precipitation or temperature, and the correlation between the climate and $N_2O$ emissions. Moreover, through comparing the spatial $N_2O$ emission map from those studies, we found that the distributions and magnitudes of emissions in the Congo, Amazon Basin, and Southeast Asia also differed significantly.

The spatial patterns of annual precipitation and temperature in this study are shown in Fig. S3. The Congo, Amazon Basin, and Southeast Asia are located in the tropics. The three regions have similar annual temperature (Fig. S2a), but have significantly different annual precipitation (Fig. S2b). In some areas in Amazon Basin and Southeast Asia, the annual precipitation was even higher than 3000 mm. In contrast, the annual precipitation in the Congo varied from 1300 to 2000 mm. In the DLEM, we explicitly considered the daily N leaching and runoff. Because of the heavy rainfall in the Amazon Basin and Southeast Asia, more N might be leached from the soil during the wet season, which could cause the lower annual $N_2O$ emissions. In addition, both denitrification and nitrification are highly affected by the soil water content. As field experiments revealed $N_2O$ or NO could be reduced into $N_2$ when soils are in saturation (Davidson et al., 2000), DLEM also represent the formation and proportion of $N_2O$ in total nitrogen oxides, considering the effect of soil moisture change. Thus, excessive soil water content during the wet season in Amazon Basin and Southeast Asia might reduce the activities of microbes, thus causing smaller amount of $N_2O$ emission.

(2) Which of the datapoints in Fig. 3 are from tropical forests and which continents are they from?
Response:

Firstly, we are sorry for the mistake in this Figure. The x-axis should be "observed $N_2O$ emission" rather than "simulated $N_2O$ emission". In the new version, we redraw this figure. In addition, we used different symbols to mark all sites in Fig. 3 to make it clearly show the locations of all 20 sites. The information of each site can be found in the supplementary material (Table S1).

(3) Is there validation support for the Congo having much higher emissions that the Amazon or SE Asia?
Response:

There were only two sites in the validation from southeast Asia (site 14) and the east coast of Austria (site 10). Unfortunately, there was no available validation to support the arguments that the Congo has much higher emissions than the Amazon Basin or Southeast Asia. However, we did find some measurements in Kim et al. (2016), which could support our estimates in the central Africa. In their study, they calculated the average $N_2O$ emission from ten observations in the Congo Basin, which was $4.2\pm1.5$ kg N ha$^{-1}$ yr$^{-1}$ and close to our estimates in this region.

(4) More discussion would be helpful to interpret the variation shown in this map, such as where agriculture was or had been, where wetlands are, and where there are hot spots other than tropical forests.
Response:

The spatial distributions of cropland and wetlands have been provided in the manuscript. Meanwhile, the emission from pre-industrial cropland was discussed in line 234–243. It is hardly to make sure the certain crop types 150 years ago. Thus, $N_2O$ emission from cropland remained quite uncertain. For the $N_2O$ emission from wetlands and peatlands, we have discussed in line 255–266. We did not include the estimate of pre-industrial wetlands or peatlands because of the uncertainty of wetland area and distribution, but it will be included in the future study. The results in the DLEM simulation indicated that where natural vegetation was, specifically the tropical forest, were hot spots for $N_2O$ emissions in the pre-industrial period. Some scattered hot spots the reviewer mentioned were also from the tropics as described below.

(5) I see a bunch of small red spots that appear to be near the Andes range, which puzzles me, but perhaps there is a good explanation. Ditto for why Northeastern Brazil, which is generally rather xeric, shows up as a hot spot. Also curious are the hot spots in southwestern China and the southeast coast of Australia.

Response:

We have noticed those "hot spots". Near the Andes range and in southwestern China, those mountains have higher altitudes and smaller amount of annual precipitation compared with the adjacent basins. Less N leaching happened in those regions. Meanwhile, both regions in the tropics that are dominant with TrBEF. Thus, it is possible that $N_2O$ emission was higher in this circumstance. In the Northeastern Brazil and the Southeast coast of Australia, both regions are along the coast. Both regions were not xeric according to the annual precipitation data used in this study. In the Northeastern Brazil, the dominant vegetation type is still TrBEF. Similarly, less N leaching and proper soil water content might cause higher amount of $N_2O$ emissions. In the east coast of Australia, anthropogenic activities contributed a large amount of N deposition in 1860 compared to other regions of Australia. Several grids with higher emissions were dominant with Temperate Broadleaf Evergreen Forest (TBEF). Meanwhile, the precipitation was generally higher along the Australian coast. Thus, higher N deposition with proper precipitation might cause this high $N_2O$ emission.

Technical Points

1. Line 41: This statement ignores that some anthropogenic emissions were already present prior to or at the beginning of the industrial revolution.

Response:

It is true that there existed anthropogenic $N_2O$ emission before 1860; however, the total amount is substantially lower than the contemporary human-induced $N_2O$ emissions. The description of anthropogenic emissions in the pre-industrial era was added, shown as "Human-induced biogenic $N_2O$ emissions are calculated by subtracting the pre-industrial emissions (Tian et al., 2016), even though a small amount of anthropogenic $N_2O$ emissions was present before 1860, which was estimated as 1.1 Tg N yr$^{-1}$ in 1850 by Syakila and Kroeze (2011) and 0.7 (0.6–0.8) Tg N yr$^{-1}$ in 1860 by Davidson (2009)."

2. Line 55: Add recent results from Prather et al. 2015.

Response:

The latest study done by Prather et al. (2015) has been added into line 56, shown as "Prather et al. (2015) provided an estimate of the pre-industrial emissions (total natural emission: 10.5 Tg N yr$^{-1}$) based on the then-most-recent model study with a corrected lifetime of 116±9 years."

3. Line 70: Change "is" to "are" because the word "data" is plural: "the data are".

Response:

It has been revised.

4. Line 178: Use estimate from Prather et al. 2015.

Response:

We have removed the section 2.4.2 and section 3.2, which described the one-box model validation of simulation results. Thus, there is no need to replace the $N_2O$ lifetime in this section. In addition, we added the comparison of the estimate in this study with the estimation by Prather et al. (2012, 2015) in the section 3.2.

5. Line 312: Consider other estimates, such as those of Voss et al. 2013.

Response:

We carefully read the paper from Voss et al. (2013) and found that their estimates were directly from the IPCC AR4 (Table 7.7 in Chapter 7). In their paper, the $N_2O$ emission from ocean was 5.5 Tg N yr$^{-1}$ because they considered the emissions (1.7 Tg N yr$^{-1}$) from "rivers, estuaries, and coastal zones" as the marine emissions. Thus, the average marine emissions are 3.8 Tg N yr$^{-1}$, as shown in Table 6.9 of Chapter 6 in the IPCC AR5.

6. Figure 2. I don't understand the units. How can these units of crop area apply to each individual pixel?

Response:

To avoid the confusion of the unit, it has been changed from "km$^2$" to "km$^2$/grid". The size of individual pixel is 0.5 degree, equivalent to around 2500 km$^2$ at the equator. Meanwhile, we have crop area fraction in each pixel (mentioned in the section 2.1 & 2.2). Then, in each grid, crop area fraction multiplying the pixel size represents the crop area. The numbers in the legend mean the cropland area in each 0.5-degree pixel.

7. Figure 3. The data used for this graph should be referenced.

Response:

All papers that used for the graph cited in the new version.

8. Figure 5. The bottom panel is all that is needed. The top panel is redundant. However, you could also add a panel of mean flux per hectare, which would be useful, because it is difficult to compare fluxes across continents when the contents have such different total areas.

Response:

We agree with the reviewer. We have removed the top panel. Instead, we added a panel of $N_2O$ emission rates per unit area (g N m$^{-2}$ yr$^{-1}$) with uncertainty ranges at continental-level in 1860, as shown in Fig. 5 (a).

9. Figure 6. The two panels are largely redundant. The pie chart could include both the percentage of the total and the estimate of Tg/yr, which would obviate the need for the upper panel. However, again, the mean flux per hectare by biome would be an interesting panel to add.

Response:

We agree with the reviewer. We have removed the top panel. Instead, we added a panel of $N_2O$ emission rates per unit area (g N m$^{-2}$ yr$^{-1}$) with uncertainty ranges at biome-scale in 1860, as shown in Fig. 6 (a). In addition, we added the biome-scale emission amounts and their uncertainty ranges into the pie chart, as shown in Fig. 6 (b).

10. Table 2. The number of significant figures shown is excessive. I suggest rounding to the nearest Gg. The uncertainties are such that any fraction of a Gg is meaningless.
Response:

Since the one-box model section has been removed, Table 1 was deleted, and "Table 2" was changed to "Table 1". The uncertainties have been removed. We added the biome- and continental-scale $N_2O$ emissions in the supplementary material (Table S2). For the mean annual $N_2O$ emissions (Tg N yr$^{-1}$) and emission rate per unit area (kg N ha$^{-1}$ yr$^{-1}$), we have listed all numbers in the Table S3. We included the revised figures as below:

[Figure]

**Fig. 2** The spatial distribution of cropland area in 1860.

[Figure]

**Fig. 3** The comparison of the DLEM-simulated $N_2O$ emissions with field observations. All sites were described in the supplementary material (Table S1).

[Figure]

**Fig. 5** Estimated N$_2$O emission rates (a) and emissions (b) with uncertainty ranges at continental-level in 1860. Solid line within each box refers to the median value of N$_2$O emission rate or amount.

[Figure]

**Fig. 6** (a) Estimated N$_2$O emission rate at biome-level in 1860 with the median value (solid line), the mean (solid dot), and the uncertainty range of emission rates from different biomes. The emission rate in the tundra was removed because of the extremely small value (less than 0.003g N m$^{-2}$ yr$^{-1}$); (b) Estimated N$_2$O emission (Tg N yr$^{-1}$) with uncertainty ranges and its percentage (%) at biome-level in 1860.

[Figure]

**Figure S2.** (a) The average annual temperature during 1901−1930; (b) The average annual precipitation during 1901−1930.

Table S1. The description of field measurements from natural vegetation in different sites.

| Number | PFT | location | | Year | References |
|--------|-----|----------|--|------|------------|
| | | Longitude | Latitude | | |
| 1 | Forest: Spruce | 11°25'E | 48°46'N | 1993-1995 | Butterbach-Bahl et al., 1998 |
| 2 | Forest: Spruce | 09°34'E | 51°46'N | 2007-2008 | Eickenscheidt and Brumme, 2012 |
| 3 | Forest: Liana canopy | 55°31'W | 3°59'S | 1998-2000 | Davidson et al., 2004 |
| 4 | Forest: Douglas-fir | 124°30'W | 44°00'N | 2007-2008 | Erickson and Perakis, 2014 |
| 5 | Grassland | 09°42'E | 51°46'N | 2008-2009 | Hoeft et al., 2012 |
| 6 | Forest | 156°14'W | 20°48'N | 2000-2001 | Holtgrieve et al., 2006 |
| 7 | Forest: Spruce &Oak | 19°57'–58'E | 47°53'N | 2002-2003 | Horváth et al., 2006 |
| 8 | Forest: Beech | 16°15'E | 48°14'N | 2002-2004 | Kitzler et al., 2006 |
| 9 | Grassland | 104°42'W | 40°50'N | 1997-2000 | Mosier et al., 2002 |
| 10 | Tropical rain forest | 145°30'E | 17°30'S | 1997-1999 | Breuer et al., 2000 |
| 11 | Tropical rain forest | 63°00'W | 10°00'S | – | Stehfest and Bouwman, 2006 |
| 12 | Savanna | 28°30'E | 24°30'S | 1994 | Scholes et al., 1997 |
| 13 | Tropical forest | 47°30'W | 3°00'S | 1987 | Luizão et al., 1989 |
| 14 | Tropical forest | 115°30'E | 2°00'S | 1998-1999 | Hadi et al., 2000 |
| 15 | Tropical forest | 84°00'W | 10°26'N | 1990-1991 | Keller and Reiners, 1994 |
| 16 | Subtropical forest | 66°00'W | 18°00'N | 1995-1996 | Erickson et al. 2001 |
| 17 | Temperate forest | 116°30'E | 39°30'N | 1997-1998 | Sun and Xu, 2001 |
| 18 | Temperate forest | 89°00'W | 43°00'N | 1979-1981 | Goodroad and Keeney, 1984 |
| 19 | Grassland | 116°04'E | 43°26'N | 1995 | Chen et al., 2000 |
| 20 | Temperate forest | 126°55'E | 41°23'N | 1994-1995 | |

Table S3 The estimated mean $N_2O$ emissions and emission rates per unit area at continental- and biome-scale with the uncertainty ranges. kg N ha$^{-1}$ yr$^{-1}$ = 0.1 g N m$^{-2}$ yr$^{-1}$

| Continental-scale | Europe | North America | South America | Southern Asia | Northern Asia | Oceania | Africa |
|---|---|---|---|---|---|---|---|
| $N_2O$ emissions (Tg N yr$^{-1}$) | 0.29 (0.21~0.40) | 0.66 (0.51~0.89) | 2.09 (1.63~2.73) | 1.16 (0.90~1.52) | 0.16 (0.11~0.26) | 0.31 (0.23~0.52) | 1.46 (1.13~1.91) |
| $N_2O$ emission rate (kg N ha$^{-1}$) | 0.31 (0.23~0.43) | 0.31 (0.24~0.42) | 1.23 (0.96~1.61) | 0.52 (0.40~0.68) | 0.13 (0.09~0.22) | 0.41 (0.31~0.69) | 0.73 (0.56~0.95) |
| Biome-scale | Boreal Forest | Tropical Forest | Temperate Forest | Shrubland | Grassland | Cropland | Tundra |
| $N_2O$ emissions (Tg N yr$^{-1}$) | 0.17 (0.10~0.25) | 4.01 (3.12~5.21) | 0.59 (0.43~0.82) | 0.82 (0.61~1.08) | 0.20 (0.15~0.25) | 0.41 (0.32~0.55) | 0.01 (0.002~0.05) |
| $N_2O$ emission rate (kg N ha$^{-1}$) | 0.17 (0.11~0.26) | 1.60 (1.25~2.09) | 0.37 (0.27~0.51) | 0.34 (0.26~0.45) | 0.2 (0.15~0.26) | 0.46 (0.36~0.61) | – |

Response:

Yes, this study was based on steady state simulation. The data sources for equilibrium run were all based on the data in 1860. Our transient run for 1860 was actually an extension of the equilibrium run. We don't have transient data before 1860 to realistically include the legacy effects from land use change, climate, etc. before 1860. The reason we ran this transient run was to avoid the abnormal fluctuations after equilibrium run, rather than capturing the legacy impacts. Fig. 4 in the manuscript is the result from equilibrium run. We made a comparison between the equilibrium and transient results for 1860 (Fig. S3). Although there were small differences for some grid cells between the two simulation results, the simulation results for the equilibrium run were similar to the transient run as a whole.

[Figure]

Fig. S3 (a) The spatial distribution of global $N_2O$ emission from the equilibrium run; (b) The spatial pattern distribution of global $N_2O$ emission from the transient run.

All changes were marked as blue in the revised manuscript.

A list of relevant changes:

1. In the **introduction** section, we added the recent results by Prather et al. (2015) (Line 55-57). Meanwhile, we included the estimation of human-induced $N_2O$ emissions in 1860 from previous studies (Line 37-41). In addition, we addressed the objectives of this study at the end of the introduction section (Line 72-77).

2. In the **methodology** section, as suggested by the reviewer, we have removed the one-box model approach.

3. In the **result and discussion** section, as suggested by the reviewer, we removed the results from the one-box approach, while we added the comparison of our study with the previous estimations based on the top-down methodology (Section 3.2). Moreover, we added one section to discuss the $N_2O$ budget in the pre-industrial era (Section 3.4).

4. In the **future research needs** section, we also made some changes, which were marked as blue (Line 353-356).

5. In the **references**, we added all missing references and marked as blue.

For tables and figure:

1. As suggested by the reviewer, we have added the site number in Table S1 and Fig. 3.

2. As suggested by the reviewers, we have revised the top panels of Fig. 5 and Fig. 6, respectively.

3. In Table 1, we have removed the uncertainty ranges for $N_2O$ emissions in each country.

4. We added Table S3 in the supplementary material, which shows the pre-industrial $N_2O$ emission amounts and rates at the continental- and biome-scale with the uncertainty ranges.

[revised manuscript text omitted]